# Human apical-out nasal organoids reveal an essential role of matrix metalloproteinases in airway epithelial differentiation

Liyue Li[1,2,8], Linyi Jiao[1,8], Danni Feng[1,8], Yizhang Yuan[1,8], Xiaoqian Yang[3], Jian Li[1,4], Dong Jiang[3], Hexin Chen[1], Qingxiang Meng[5], Ruchong Chen[6], Bixing Fang[1], Xuenong Zou[3], Zhenhua Luo ●[3], Xiaoyan Ye[1,4], Yue Hong ●[7], Chun Liu ●[3] ✉ & Chunwei Li ●[1] ✉

Extracellular matrix (ECM) assembly/disassembly is a critical regulator for airway epithelial development and remodeling. Airway organoid is widely used in respiratory research, yet there is limited study to indicate the roles and mechanisms of ECM organization in epithelial growth and differentiation by using in vitro organoid system. Moreover, most of current Matrigel-based airway organoids are in basal-out orientation where accessing the apical surface is challenging. We present a human apical-out airway organoid using a biochemically defined hybrid hydrogel system. During human nasal epithelial progenitor cells (hNEPCs) differentiation, the gel gradually degrade, leading to the organoid apical surfaces facing outward. The expression and activity of ECM-degrading enzymes, matrix metalloproteinases (MMP7, MMP9, MMP10 and MMP13) increases during organoid differentiation, where inhibition of MMPs significantly suppresses the normal ciliation, resulting in increased goblet cell proportion. Moreover, a decrease of MMPs is found in goblet cell hyperplastic epithelium in inflammatory mucosa. This system reveals essential roles of epithelial-derived MMPs on epithelial cell fate determination, and provides an applicable platform enabling further study for ECM in regulating airway development in health and diseases.

The airway epithelium is critical for maintaining the homeostasis of respiratory mucosa. Dysregulation of epithelia leads to abnormal remodeling that can initiate inflammation in airway mucosa[1–3]. The respiratory epithelium, ranging from sinonasal cavity to large bronchioles, is called mucociliary epithelium that mainly contains basal, secretory (mainly goblet cells), ciliated cells and extracellular matrix (ECM). Among them, basal cells are functioning as progenitors of the multiciliated and secretory cell populations[4,5]. The development of airway epithelium involves a series of complex alterations in the cellular microenvironment, where the interaction between progenitor

[1]Department of Otolaryngology, Department of Allergy, Guangzhou Key Laboratory of Otorhinolaryngology, The First Affiliated Hospital, Sun Yat-sen University, Guangzhou, China. [2]Department of Otorhinolaryngology, Union Hospital, Tongji Medical College, Huazhong University of Science and Technology, Wuhan, China. [3]Precision Medicine Institute, Guangdong Provincial Key Laboratory of Orthopedics and Traumatology, The First Affiliated Hospital of Sun Yat-sen University, Guangzhou, China. [4]Guangxi Hospital Division of The First Affiliated Hospital, Sun Yat-sen University, Nanning, China. [5]Department of Otorhinolaryngology Head and Neck Surgery, Guangzhou First People's Hospital, Guangzhou, China. [6]State Key Laboratory of Respiratory Disease, National Clinical Research Center for Respiratory Disease, Guangzhou Institute of Respiratory Health, Department of Allergy and Clinical Immunology, Guangzhou Medical University, Guangzhou, China. [7]School of Life Sciences, Hainan University, Haikou, China. [8]These authors contributed equally: Liyue Li, Linyi Jiao, Danni Feng, Yizhang Yuan. ✉e-mail: liuch393@mail.sysu.edu.cn; hi_chunwei@aliyun.com

cells and ECM has been demonstrated to play a key role in epithelium formation, as well as disease occurrence[6]. Indeed, ECM (especially collagens) is essential for controlling cell behavior during tissue formation in many organ systems[7–10]; and change of ECM assembly and disassembly properties is a critical feature for airway epithelial development, while imbalance of ECM organization can lead to abnormal epithelial remodeling[6,11,12]. However, the mechanisms of how epithelial progenitor cells interact with ECM during airway development remain unclear, which is partially due to the lack of a biomimetic model that can recapitulate the development of progenitor cells within ECM in vitro.

Airway organoids have recently emerged as a promising experimental tool to model the human respiratory tract and the organoids have been successfully used to address outstanding questions in respiratory biology[13–16]. Generally, lung or airway organoids refer to self-assembling structures of epithelial cells cultured in 3D. And the cellular origins of organoids include basal progenitor cells from adult tissues, embryonic stem cells, and induced pluripotent stem cells[17]. Most of current organoid models are generated using 'Matrigel', which is a commercialized hydrogel extracted from a murine tumor[18]. However, very few studies have explored the interactions between cells and ECM using Matrigel[18]. This is mainly due to the relative stable hydrogel properties caused by inherent laminin and type IV collagen that limit the capability of cells to remodel ECM. The degradation of Matrigel during cell culture is a slow process which can last as long as a few months, and the lack of type I collagen largely restricted the migration of cells. Therefore, instead of growing outward by degrading the surrounding gel and migration, cells in Matrigel tend to grow more inwards to form spheroids, leading to a basal-out orientation as reported in most of current airway organoid models[14,19], i.e., the apical surface (ciliated cells or goblet cells) of the epithelium is enclosed within the spheroid. Such orientation pattern may not fully recapitulate the in vivo epithelial structure character, as the apical surface of airway epithelium should face to atmosphere and readily interact with luminal contents. Moreover, the current basal-out models lead to difficulties in studying the epithelial host-stimulant interactions, as the apical layer is hard to access by pathogens. Although besides the Matrigel based models, there has also been commercialized culture system that can enable airway spheroids in an apical-out manner, the lack of 3D hydrogel environment made it inappropriate to study the cell-ECM interaction[20,21]. Therefore, there is a need to develop a simple, chemically defined, highly tunable, and tissue-specific hydrogel alternatives, in order to study the role of ECM during airway epithelial development and remodeling in a more physiologically-relevant manner.

We previously have shown the collagen/alginate hydrogel as a biomimetic ECM material for cancer cell proliferation and migration[22,23], where collagen network were shown to be interactively remodeled by stromal cells. Here, we further defined the gel recipe by adding hyaluronic acid, which is an important ECM component for moisture reservation during epithelial development; hence, such hybrid hydrogel is composed of collagen type I, alginate and hyaluronic acid (CAH gel). Different from the existing organoid systems, by incorporating human nasal epithelial progenitor cell (hNEPC) culture technique in CAH gel, we have successfully produced an apical-out airway organoid culture system that maintains the ability of organoids to proliferate and grow outward into branched structures as differentiating into ciliated cells or goblet cells. We found that the expression and activity of matrix metalloproteinases (MMPs) was elevated during organoid differentiation accompanied with gel degradation on the surface; while inhibition of MMP activity significantly suppressed the ciliation in organoids but promoted the increase of mucus secreting cell. Consistent findings were found in inflammatory nasal mucosa, showing that a decrease of MMP7, MMP9, MMP10, and MMP13 expressions in the epithelium with goblet cell hyperplasia

compared to normal ciliated epithelium. The above findings indicate a broad application of our system in studying the mechanisms underlying airway epithelial development during normal and disease stages in a clinical relevant manner.

## Results

### Establishment of CAH gel based human apical-out nasal organoids (hANOs)

We have collected nasal mucosa biopsies from patients with chronic rhinosinusitis under functional endoscopic surgery. Epithelial cells were isolated through mechanical disruption and enzymatic dissociation of tissue samples. hNEPCs were grown in a feeder-based system as referred by previous airway epithelial stem cell studies[24–26]. Based on our experience with biomaterials in 3D in vitro cell culture[22,23], we designed a hybrid CAH gel containing collagen type I (C), alginate (A), and hyaluronic acid (H), in which the collagen and hyaluronic acid represent the common ECM components in airway mucosal tissues[27]. We then customized the CAH hydrogel combined with hNEPC culture conditions and established a nasal organoid system: (1) hNEPCs in CAH gel were proliferating and extending in random directions, then forming into 3D branched structures from day 5 to day 10 in epithelial growth media (Fig. 1A, Supplementary Fig. 1 and Supplementary Movie 1); (2). Once the organoids reached high density, the culture condition was changed to the differentiation media, and the nasal organoids gradually transformed into a spheroid-type morphology from day 11 to day 17 (Fig. 1A, Supplementary Fig. 1 and Supplementary Movie 2); (3) The organoids further aggregated and partially merged into bigger spheroids from day 18 to day 24 (Fig. 1A, Supplementary Fig. 1 and Supplementary Movie 3), and abundant beating cilia appeared in the organoids (Supplementary Movie 4). We have successfully generated 20 lines of differentiated apical-out nasal organoids from 20 different donors, indicating the robustness and reproducibility of the current system.

Compared to those Matrigel based apical-in lung organoids that often maintain a clear hollow lumen (Figures as shown in literature report[19]), the current differentiated nasal organoids show a solid cellular sphere and lack a discernible central lumen (Fig. 1A). This pattern under bright field microscopy is also similar with the recent report showing solid spheres in Matrigel based apical-in nasal organoids[28]. The 3D immunofluorescence images captured by multiphoton microscope showed that during proliferation stage the major cell type belongs to basal cells (p63+ cells) (Fig. 1A). During differentiation stage, goblet cells (MUC5AC+ secreting cells) appeared first on around day 17, and ciliated cells (βIV-tubulin+ or FOXJ1+) markedly present on around day 24; at this stage p63+ basal or progenitor cells decreased (Fig. 1A, B). In differentiated stage, the apical tight junctions also formed, as indicated by the tight junction protein (ZO-1) (Fig. 1B, Supplementary Fig. 2); while other epithelial cell type like club cells was also identified in the organoids (Supplementary Fig. 2). The z-plane view of the staining images displayed a few of small lumens inside the differentiated organoid. Cilia structure protein (βIV-tubulin) staining of the organoid border clearly revealed that the apical surfaces of organoids faced outward, and hence we called them human apical-out nasal organoids (hANOs). Moreover, the multiphoton microscopic images also indicated that fusion of organoids during late differentiation stage was mediated via a basal-to-basal docking pattern, but apical sites with ciliated cells were not merged (Supplementary Fig. 3).

Interestingly, we found that the CAH gel can maintain its integrity during the proliferation stage (until Day 10); while following the progression of differentiation, the gel was gradually degraded (Fig. 1C). By using fluorescent probes of Collagen Hybridizing Peptides (F-CHP) that specifically bind to denatured collagen strands (i.e., degraded collagen) but do not bind to intact collagens[29], we showed that the signals of F-CHP were significantly enhanced in differentiated hANOs (Day 24) when compared to proliferated hANOs (Day 10) (Fig. 1C, D).

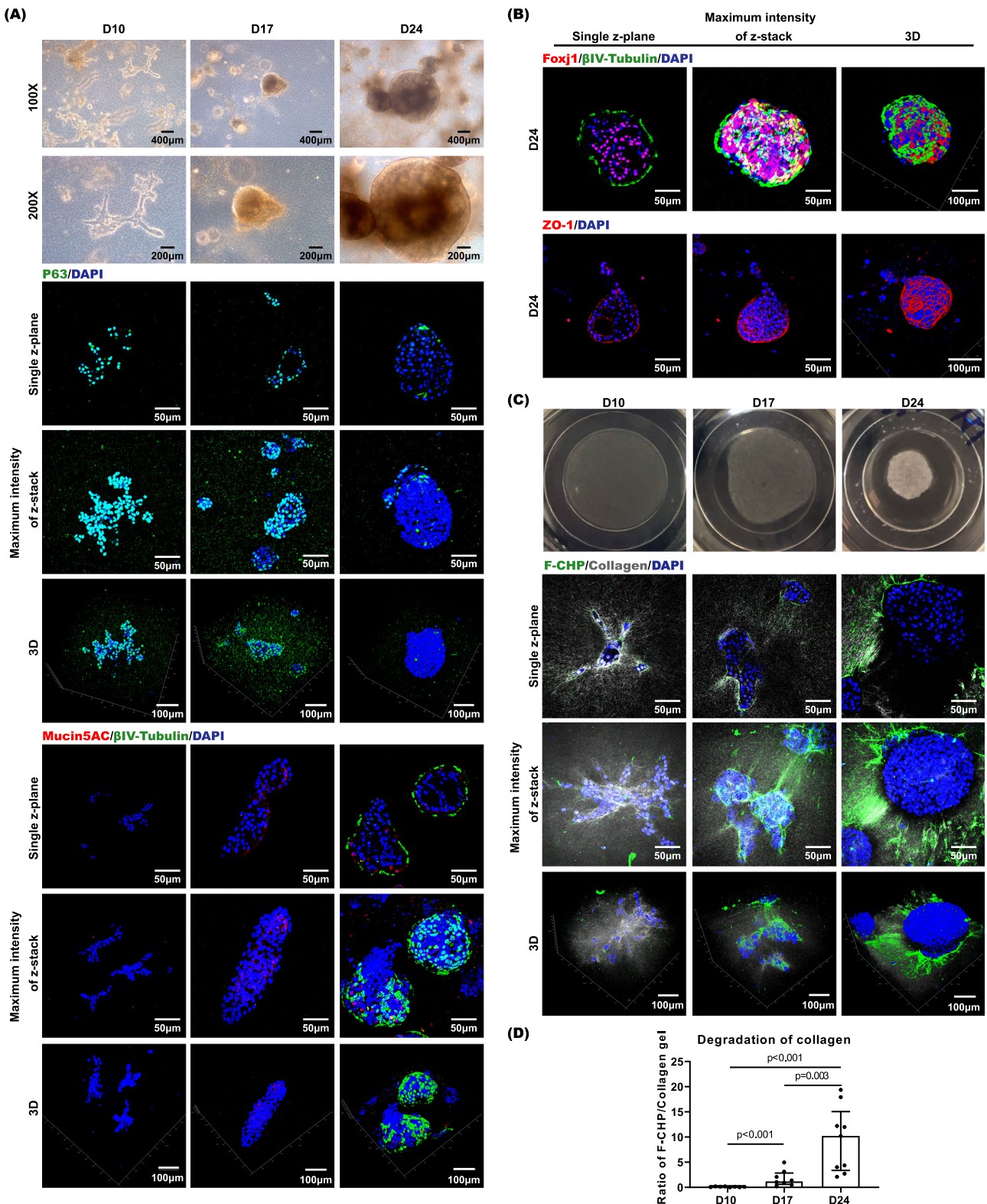

**Fig. 1 | Characterization of hANOs during epithelial differentiation.**
**A**, **B** Representative pictures of hANOs morphology (by bright field microscopy, at 100x and 200x magnification, respectively) and candidate markers (p63, βIV-tubulin, MUC5AC, FOXJ1, and ZO-1) of hANO development (by multiphoton microscopy) at different stages of hANO differentiation (D10, D17, and D24). Time lapse microscopy videos for hANOs in different differentiation stages shown in Supplementary Movie 1 to 3; Cilia beating patterns of hANOs shown in Supplementary Movie 4. **C** Gel degradation and F-CHP at different stages of hANO differentiation (D10, D17, and D24). **D** Collagen degradation was analyzed by quantifying the ratio of F-CFP/Collagen gel signals; the Mann–Whitney test was used in comparison analysis; data present the median with interquartile range, $n = 8$ experiments per condition; the experiment was independently performed in three organoid lines from three different donors; ns, not significant.

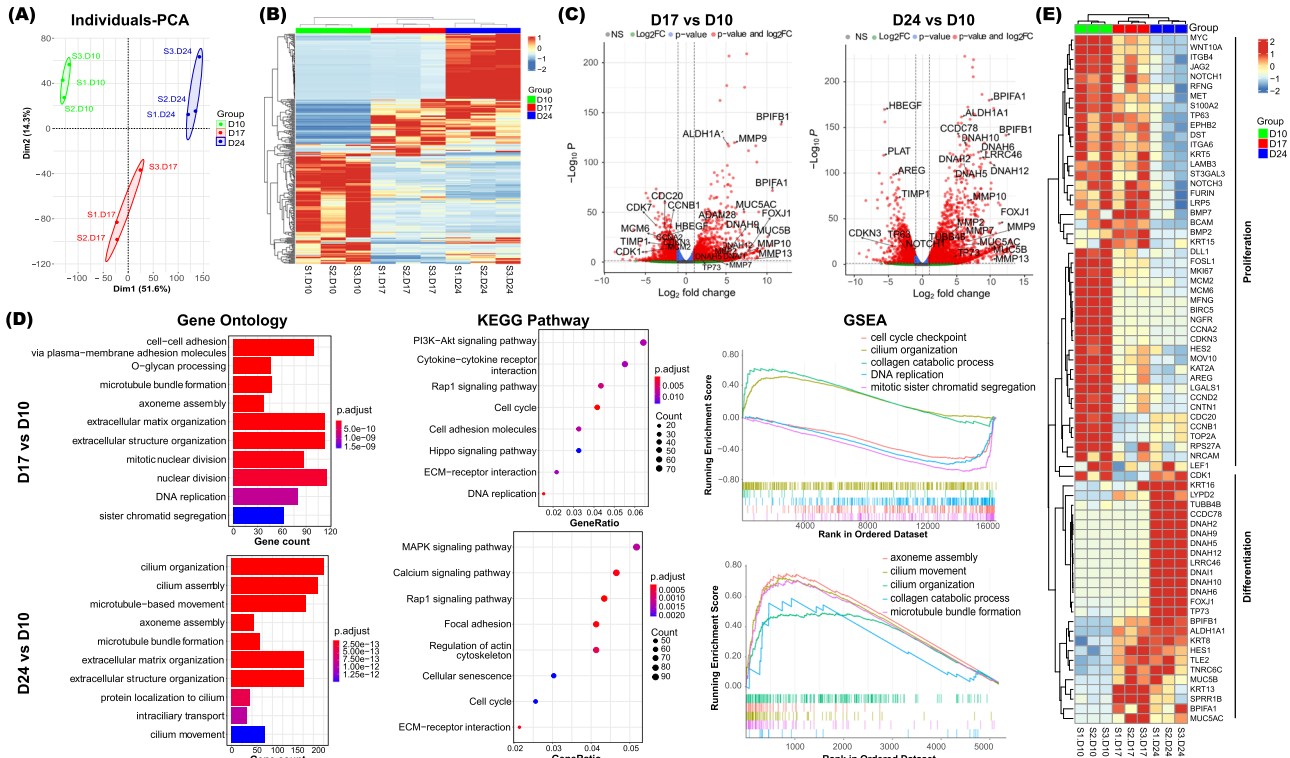

**Fig. 2 | Molecular characteristics of hANO development. A**, **B** PCA plots show distinct molecular profiles during hANOs differentiation. **C** Volcano plots show the differentially expressed genes in differentiating (D17) or differentiated (D24) cells versus proliferated cells (D10). Differentially expressed genes in different stages of hANOs shown in Supplementary Data 1. **D** Biological process, KEGG analysis, and GSEA analysis demonstrate the relevant functional terms in different differentiation stages of hANOs. **E** Heatmap demonstrates key genes associated with airway epithelial proliferation and differentiation based on RNAseq analysis. The values in heatmaps are the log2-scaled RNA-seq read counts. $n = 3$ experiment per condition; the experiment was independently performed in three organoid lines from three different donors.

These findings indicate that degradation of the gel may generate some empty space, in which the apical layers of organoids can orientate towards the lumen.

Collectively, we successfully developed apical-out upper airway organoids with mucociliary differentiation in a Matrigel alternative and degradable gel system in which this system could recapitulate the 3D structural characteristics and the cellular composition of human nasal epithelium.

### hANO establishment by collection of epithelial cells from nasal mucosa using brushing technique

To test the applicability of CAH gel culture system to other sources of epithelial cells, we collected nasal epithelial cells from the middle turbinate of CRS patients ($n = 2$) and control subjects ($n = 2$) using a nasal brushing technique. The cells from mucosal surface were obtained by a brush and then the total epithelial cells (CD45-CD90-CD142+) were sorted via flow cytometry. The isolated epithelial cells (in an optimal density $5 \times 10^4$ per well) were directly cultured in the hybrid hydrogel system without pre-2D expansion procedure. The cell units grew in a branched structure at proliferation stage; during differentiation stage, the gel is also degraded, and the organoids became spheroid-like morphology with beating cilia facing outward (Supplementary Fig. 4). Thus, we confirm the hANOs can be generated by using the epithelial cells collected by minimally invasive approach.

### Molecular characteristics and features of hANOs during differentiation progression

To examine changes in genes expression profiles during the development of hANOs, RNA-seq analysis was performed at three different time points: the end of proliferation (Day 10), differentiating stage

(Day 17), and differentiated stage (Day 24). Based on unsupervised principal component analysis (PCA) and Heatmap analysis, the individual samples were separated into three distinct groups that corresponded with the three differentiation stages (Fig. 2A, B). Differentially expressed genes (DEGs) in hANOs at the three differentiation steps were analyzed. There were 3251 up-regulated gens and 2082 down-regulated genes in hANOs on Day 24 compared with those on Day 10 (Fig. 2C, Supplementary Data 1). To gain an understanding of the biological functions occurring during the differentiation of hANOs, we performed gene ontology (GO) analysis, gene set enrichment analysis (GSEA), and pathway analysis. Of the top 10 terms that define relevant biological processes, most were related to ciliogenesis (e.g., cilium organization, cilium assembly, microtubule-based movement etc.) and cell cycle regulation (e.g., nuclear division, DNA replication, and chromosome segregation, etc.), while two categories were associated with ECM assembly (extracellular matrix organization, extracellular structure organization) (Fig. 2D). KEGG analysis revealed that pathways related to cell cycle, ECM-receptor interaction, Rap1 signaling were involved in the entire differentiation process, while MAPK signaling and Calcium signaling were associated with the stages of full differentiation (Fig. 2D). GSEA results demonstrated that during early differentiation (Day 17 vs. Day 10), the functional activities of mitotic sister chromatid segregation, cell cycle checkpoint and cell cycle phase transition were significantly decreased (Fig. 2D); while when hANOs were fully differentiated, activities with cilium movement, axoneme assembly, and cilium organization were significantly enhanced (Fig. 2D). Interestingly, the collagen-catabolic process that involves MMP family members was significantly activated during the differentiation progression (Fig. 2D). Finally, a decrease of cell proliferation associated genes (e.g., *TP63*, *NOTCH1*, *MKI67*, and *MYC*) but

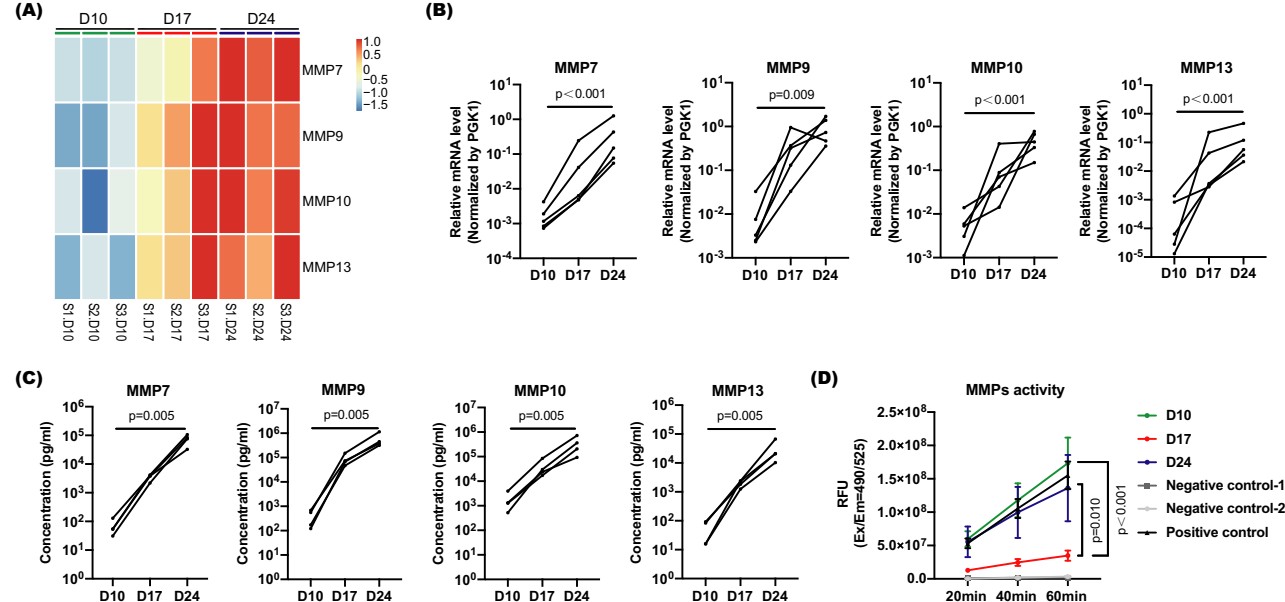

**Fig. 3 | Expression and activity of MMPs during hANO development. A** Heatmap shows expression levels of MMP7, MMP9, MMP10 and MMP13 based on RNA-seq data. **B** mRNA levels of *MMP7*, *MMP9*, *MMP10*, and *MMP13* were analyzed in hANOs at D10, D17 and D24 by qPCR assay; the experiment was independently performed in five organoid lines from five different donors. **C** Protein levels of MMP7, MMP9, MMP10, and MMP13 were analyzed in hANOs at D10, D17 and D24 by Luminex assay; the experiment was independently performed in four organoid lines from four different donors. **D** Pan-MMP activity was analyzed in hANOs at D10, D17, and D24, Pan-MMP activity was analyzed in hANOs at D10, D17, and D24; NC-1 and NC-2 were negative controls in which the organoids treated with culture medium and assay buffer, respectively; while PC was the positive control in which the organoids treated with APMA pre-treated MMP9 recombinant protein; RFU relative fluorescence units; data present the mean with SEM, *n* = 5 experiments per condition; the experiment was independently performed in five organoid lines from five different donors. The Friedman test was used in comparison analysis in (**B**) and (**C**); the 2way ANOVA with Tukey's multiple comparison test was performed in (**D**).

an increase of genes related to airway epithelial cell differentiation (e.g., *FOXJ1*, *TP73*, *ALDH1A1*, and *KRT8*) was observed following organoid differentiation (Fig. 2E). Taken together, the RNA-seq results suggested the role of cell cycle and ciliogenesis during differentiation stages of hANOs. Moreover, the data suggested that the functional activities related to ECM assembly and collagen catabolic process may contribute to the degradation phenotype of CAH gel during epithelial differentiation.

**Comparisons of molecular characteristics of nasal epithelial differentiation in hANOs versus air-liquid interface (ALI) model**
ALI is a well-known model to induce airway epithelial differentiation. By using the same hNEPCs culture methods, progenitor cells were transferred to a transwell system and then differentiated into mucociliated cell types. Similar to hANO setting, RNA-seq experiment was performed at the end of proliferation (Day 7), differentiating stage (Day 14), and differentiated stage (Day 21). There were 5434 and 7801 DEGs at differentiating and differentiated stage respectively as compared to proliferation stage; and the GO analysis present significant ciliogenesis related functions during differentiation progression (Supplementary Fig. 5). We then compared the similarity of the epithelial development profiles between hANO and ALI models. The results showed that at differentiating and differentiated stages the overlapping DEGs of these two models were 4541 (82% among the total DEGs) and 7025 (88% among the total DEGs), respectively (Supplementary Fig. 5). Importantly, both organoid and ALI systems show an alteration of ECM organization related functional terms during epithelial differentiation; gropus of ECM related genes were significantly changed in a similar trend at proliferation and differentiation status (Supplementary Fig. 5). The above findings show similar molecular characteristics of the epithelial cells between current hANOs and traditional ALI model, suggesting that the presence of biological functions (like ECM assembly/disassembly) is not an artifact of the culture in this CAH gel system.

**Increased expression and activity of MMPs (MMP7, MMP9, MMP10, and MMP13) in hANOs during differentiation**
MMPs are zinc-dependent proteases that are responsible for degrading ECM components (including collagens) in many tissues[6,9,30]. As collagen assembly and disassembly systems can be modulated by different MMP family members, we assumed that the degradation of the current CAH gel system could be attributed to certain MMPs produced by nasal epithelial cells during the differentiation process.

Based on the RNA-seq analysis, we identified potential MMP candidates by examination of the genes in categories related to ECM organization. We screened MMP genes based on their changes in expression levels with high statistical significance between differentiation (D17 and D24) and proliferation (D10) stages (Supplementary Table 1); and then shortlisted several MMP members with significant up-regulation trend during the progression of differentiation, including MMP7, MMP9, MMP10, and MMP13 (Fig. 3A). We further validated their mRNA and protein expression levels. mRNA levels of *MMP7*, *MMP9*, *MMP10*, and *MMP13* were significantly up-regulated in differentiating (Day 17) and differentiated (Day 24) stages as compared to the end of proliferation stage (Day 10) (Fig. 3B). By using Luminex assay, the production of MMP7, MMP9, MMP10, and MMP13 proteins was also significantly increased in the supernatants of hANOs during the differentiation process (Day 24 or Day 17 versus Day 10) (Fig. 3C). In addition, the protease activity of total MMPs was analyzed, and hANOs in differentiating status (Day 17) had a significantly lower proteolytic activity as compared to differentiated hANOs (Day 24) (Fig. 3D). The above results indicate that epithelia-derived MMP7, MMP9, MMP10, and MMP13 are involved in the degradation of collagen gel during differentiation of hANOs.

MMP genes expression was also validated in other airway epithelial culture systems, including ALI and Matrigel model. Following differentiation progression, mRNA levels of *MMP7*, *MMP9*, *MMP10*, and *MMP13* were significantly up-regulated in ALI; while except MMP7, the other three MMP genes showed an up-regulation trend in Matrigel

model (Supplementary Fig. 6). These findings indicate that production of MMP genes in airway epithelial cells during differentiation is a general phenomenon and it is not induced by the current hydrogel condition.

## Epithelial-derived MMP activity is required for normal muco-ciliary differentiation of hANOs

To determine whether hANO differentiation was dependent on MMP protease activity, organoids were cultured in the presence of MMP inhibitor (actinonin) from Day 10 (i.e., the end of proliferation culture). There was a significant decrease of MMP activity in hANOs treated with actinonin during the differentiating (Day 17) and differentiated (Day 24) stages (Fig. 4A). Based on the microscopic observation, we found that the organoids treated MMP inhibitor showed less spheroid-type morphology (i.e., maintaining the branched structure) during the differentiating period (from Day 11 to Day 17) as compared to the untreated organoids (Supplementary Fig. 7); on Day 24, although both actinonin treated and untreated organoids displayed similar size of merged spheroids, interestingly, fewer beating cilia were observed in actinonin treated organoids (Supplementary Fig. 7 and Supplementary Movie 5).

By evaluating the percentages of ciliated cells and goblet cells as well as the expression levels of their respective specific candidates (Foxj1 and MUC5AC), hANOs treated with actinonin showed a significant reduction in cilia formation; however, these hANOs showed enhanced goblet cell proportion (Fig. 4B–D). With regard to the gel degradation, organoids treated with MMP inhibitor appeared to show a delayed degradation of collagen compared to the untreated organoids, as seen on the gels and CHP data (Fig. 4E–G). The above results revealed that inhibition of MMP activity could reduce the ciliation of organoids, and lead to an abnormal differentiation phenotype, i.e., a relative increase in the number of goblet cells.

RNA-seq analysis was utilized to further characterize the molecular profiles in hANOs treated with MMP inhibitor versus those that were untreated. The PCA plots and Heatmap displayed that treated and untreated hANOs were separated into different clusters (Fig. 5A, B). There were 468 differentiated genes (129 up-regulated, 339 down-regulated) and 1766 differentiated genes (438 up-regulated, 1328 down-regulated) in hANOs treated with MMP inhibitor compared to the untreated ones at Day 17 and Day 24, respectively (Fig. 5C, Supplementary Data 2). GO analysis showed that these genes were most enriched in ciliogenesis- related functions categories in actinonin treated hANOs versus those untreated (Fig. 5D). GSEA results indicated that when comparing MMP inhibitor-treated organoids versus untreated organoids, the cilium assembly activity was decreased and the activities of extracellular matrix assembly was enhanced; moreover, the functions of host defense response and Wnt signaling pathway were activated in actinonin-treated organoids compared to untreated organoids at differentiated stage (Day 24) (Fig. 5D). In actinonin-treated versus untreated hANOs, there was a significant downregulation of signature ciliogenesis-related genes such as *FOXJ1*, *DNAAF1*, *DNAH5*, *DNAH12* and *TUBB4B*, but an upregulation of genes contributing to goblet cell differentiation or mucus function were significantly up-regulated, such as *SPDEF*, *IL-13*, *FOXA3*, *MUC5AC* and *MUC5B* (Fig. 5E). Therefore, the molecular characteristics of hANOs treated with MMP inhibitor suggest a suppression of ciliogenesis but also an enhancement of the mucus-secreting cells. These data are consistent with the expected change of cellular phenotypes.

## The ciliation process of airway epithelium is dependent on MMP activity in ALI and apical-in organoid model

To validate the above findings of hANOs, we further investigate the role of MMPs on ciliogenesis of nasal epithelium in ALI and Matrigel system. By using hNEPCs in Matrigel culture conditions based on the manufacture's protocols, differentiated nasal organoids were

successfully established. Under bright field microscope view, the organoids maintained solid cellular spheres without a clear central lumen throughout proliferation and differentiation stages, in which this pattern was similar to those reported in the literature[28]. The gel was intact without any degradation. The cross-sectional images of organoids showed an apical-in polarity with the ciliated cells oriented inwardly (Supplementary Fig. 8). When applying MMP inhibitor in the cells in ALI and Matrigel model from the initiation of differentiation, the proportion of ciliated cells as well as the signature gene *FOXJ1* was significantly decreased compared to the untreated cells; while the percentages of goblet cells appear to be unchanging in the actinonin treated model versus the untreated model (Supplementary Fig. 8). Taking together, these results are in consistent with the presented hydrogel-based apical-out organoids, confirming the important roles of MMPs in airway ciliation process, and highlighting the normal airway epithelial differentiation is dependent on MMP activity.

## Down-regulation of MMPs in goblet cell hyperplasia in chronic inflammatory nasal mucosa

To examine whether epithelial-derived MMPs are involved in epithelial remodeling in airway mucosa, we used mucosa from nasal polyps to evaluate the expression levels of MMP7, MMP9, MMP10, and MMP13 in normal epithelium (from control subjects) versus epithelium with goblet cell hyperplasia (from CRS patients). Masson staining results displayed that the basement membrane beneath epithelium that is mainly composed of collagens was thicker in goblet cell hyperplastic epithelium than in the normal epithelium (Fig. 6), indicating that an increase of collagen deposition may be related to abnormal epithelial remodeling. Staining of MMP proteins demonstrated that MMP7, MMP9, MMP10, and MMP13 were mainly located in ciliated cells and some basal cells in normal epithelium (Fig. 6); while in epithelium with goblet cell metaplasia, expression levels of MMP7, MMP9 and MMP13 were significantly down-regulated as compared to normal epithelium covered by ciliated cells (Fig. 6). It appeared that MMP10 universally expressed in both normal and metaplastic epithelium (Fig. 6). Other than epithelial region, we found that these MMP proteins also localized in certain types of immune cells in sub-epithelial region (Fig. 6). Taken together, epithelium-derived MMP expression patterns in in vivo tissues showed similar changes to those observed in hANO models, and supported the in vitro findings that up-regulation of MMP (MMP7, MMP9 and MMP13) expression is associated with airway epithelial ciliogenesis, but that inhibition of MMPs may contribute to an abnormal differentiation pattern, namely an increase of the number of goblet cells.

## Specific MMPs are essential for ciliated cell differentiation of hANOs

We further clarify that the mucociliary development of hANOs might be attributed to particular MMP. Based on commercial availability of inhibitory compounds or neutralizing antibodies for specific MMPs, we selected inhibitors for MMP9 and MMP13, respectively, and neutralizing antibodies for MMP7 and MMP10, respectively. The differentiation phenotypes of hANOs were compared in the organoids in presence with MMP9 inhibitor, MMP13 inhibitor, MMP7 neutralizing antibody and MMP10 neutralizing antibody versus untreated organoids. The extent of CAH gel degradation in organoids treated with MMP13 inhibitor was comparable to those seen in untreated organoids; while the treatment with MMP9 inhibitor and neutralizing antibody for MMP7 or MMP10 showed less gel degradation than seen in untreated organoids (Fig. 7A). Immunofluorescent staining displayed that the treatment with MMP9 inhibitor and neutralizing antibody for MMP7 or MMP10 significantly reduced the ability of organoids to differentiate into ciliated cells as compared to those untreated or organoids treated with MMP13 inhibitor (Fig. 7B–D). When comparing goblet cells in

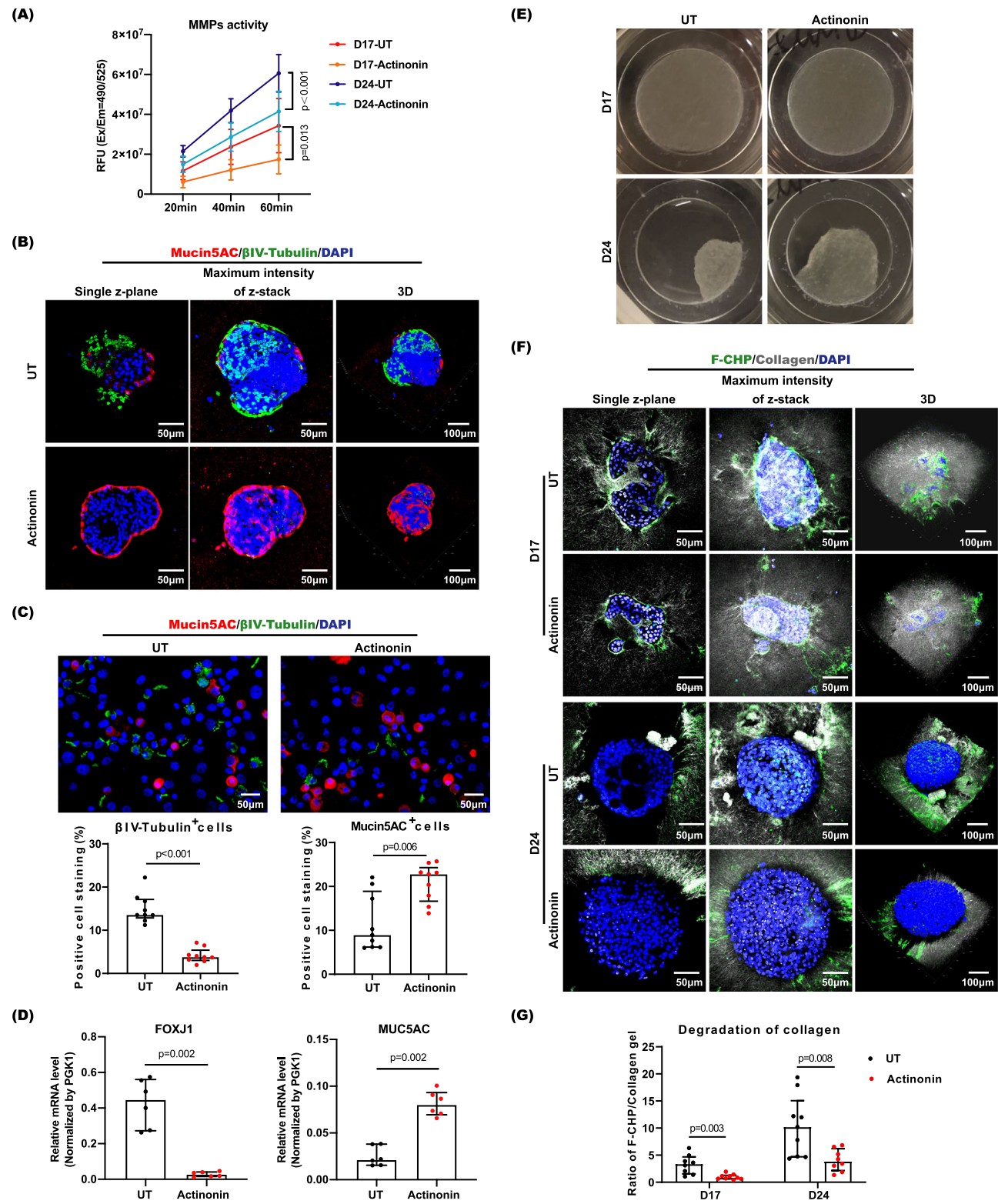

untreated organoids, organoids treated with MMP9 inhibitor showed an increase of goblet cell percentage, but organoids treated with MMP13 inhibitor, neutralizing antibody for MMP7 or MMP10 didn't have significant effect on goblet cell differentiation (Fig. 7B–D). Quantitative PCR also showed a down-regulation of ciliated cell signature genes (FOXJ1) in MMP9 inhibitor-treated organoids, MMP7 or MMP10 neutralizing antibody treated organoids when compared to untreated controls; while up-regulation of mucin genes (MUC5AC) produced by goblet cells was only found in organoids treated with

MMP9 inhibitor versus untreated organoids (Fig. 7E). We further tested the application of siRNA assay in hANO model by applying siRNA for MMP9 in organoids during the differentiation progression. The knockdown efficiency of MMP9 siRNA in differentiated organoids was about 33%; while we still found a delay of gel degradation and reduction of ciliated cell proportion in organoids treated with siRNA-MMP9 compared to those with negative control siRNA (Supplementary Fig. 9). The above findings suggest that the normal mucociliary differentiation especially ciliation of hANOs could be

**Fig. 4 | Characterization of hANOs treated with MMP inhibitor. A** MMP activity was measured in actinonin treated and untreated hANOs, data present the mean with SEM, *n* = 4 experiments per condition; the experiment was independently performed in three organoid lines from three different donors. **B** Representative pictures of candidate cellular markers (βIV-tubulin and MUC5AC) of hANO (by multiphoton microscopy and immunofluorescent microscopy) in differentiated hANOs (D24) treated with actinonin versus untreated organoids. **C, D** βIV-tubulin⁺ cells, MUC5AC⁺ cells and mRNA expression of *FOXJ1* and *MUC5AC* were compared in actinonin-treated to untreated hANOs; for positive cell analysis, data present the median with interquartile, *n* = 9 experiments per condition; for qPCR analysis, data

present the mean with SD, *n* = 6 experiments per condition; the above two experiments were independently performed in three organoid lines from three different donors. **E, F** Gel degradation and F-CHP in differentiating (D17) and differentiated (D24) hANOs treated with actinonin versus untreated organoids. **G** Collagen degradation was analyzed by quantifying the ratio of F-CFP/Collagen gel signals, data present the median with interquartile, *n* = 9 experiments per condition; the experiment was independently performed in three organoid lines from three different donors. The 2way ANOVA with Tukey's multiple comparison test was performed in (**A**); the Mann–Whitney test was used in comparison analysis in (**C**), (**D**) and (**G**); ns not significant.

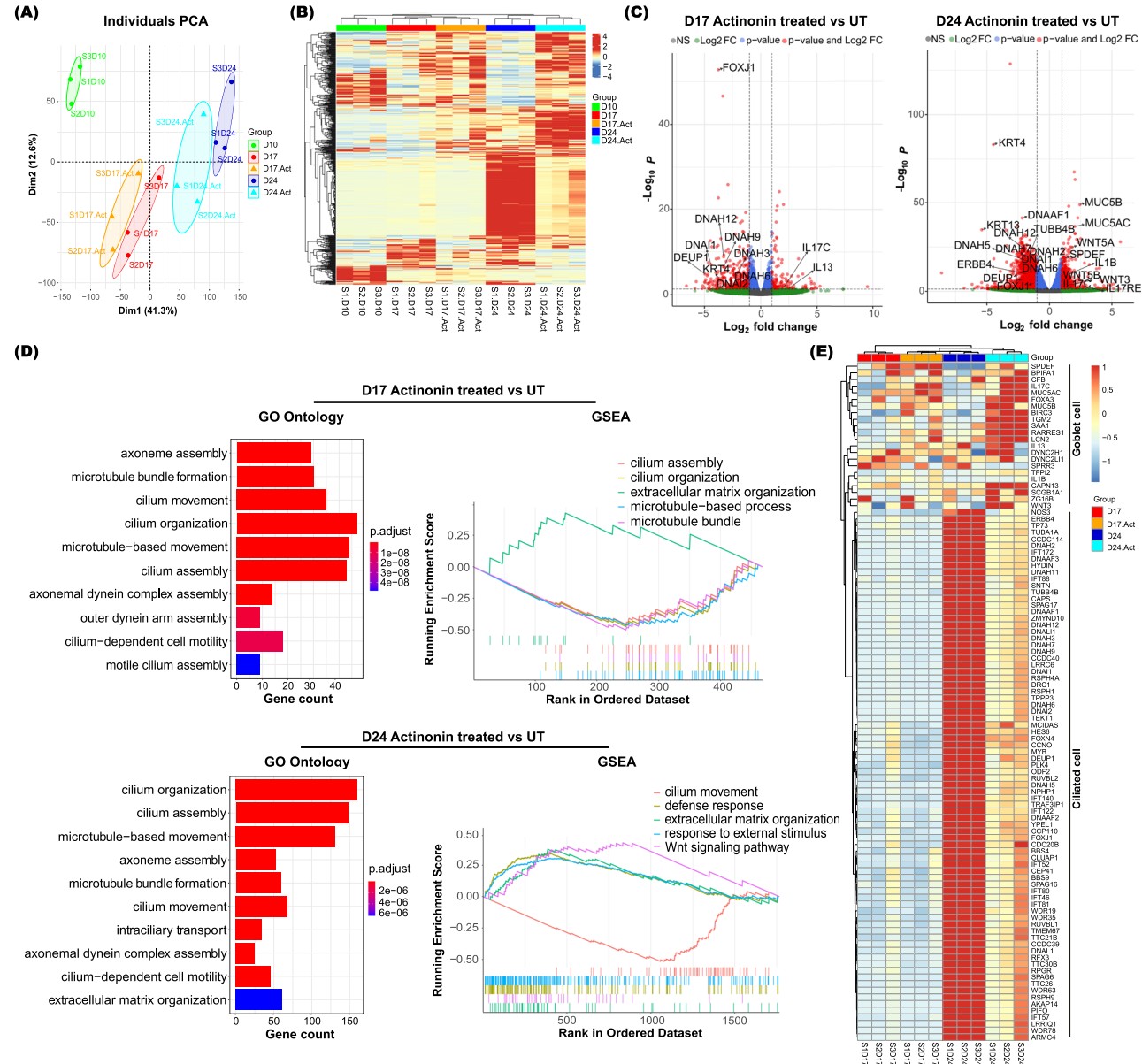

**Fig. 5 | Molecular characteristics of hANOs treated with actinonin. A, B** PCA plots and Heatmap show distinct molecular profiles in actinonin-treated versus untreated hANOs; the values in heatmaps are the log2-scaled RNA-seq read counts. **C** Volcano plots show the differentially expressed genes in actinonin-treated versus untreated hANOs. Differentially expressed genes in actinonin-treated versus untreated hANOs shown in Supplementary Data 2. **D** Biological process and GSEA

analysis demonstrates the changes of key relevant functional terms in actinonin-treated versus untreated hANOs. **E** Heatmap demonstrats key genes associated with ciliated cells and goblet cells in hANOs treated with and without actinonin during epithelial differentiation. *n* = 3 experiments per condition; the experiment was independently performed in three organoid lines from three different donors.

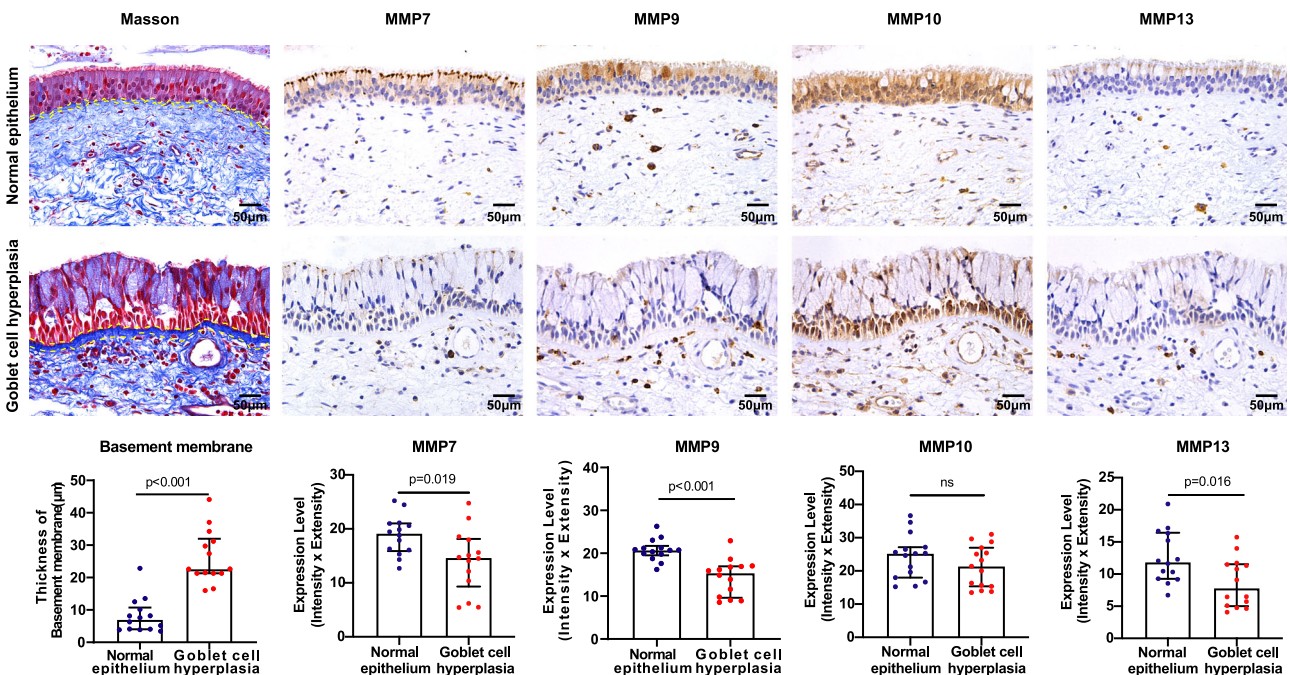

**Fig. 6 | Evaluation of basement membrane and MMP expression in normal epithelium and epithelium with goblet cell hyperplasia in nasal mucosa.** Representative pictures of MMP7, MMP9, MMP10, and MMP13 in normal and remodeled nasal epithelium. Thickness of basement membrane and MMP protein levels were compared between normal and goblet cell hyperplastic epithelium. Data present the median with interquartile, $n = 14$ experiments per group; the experiment was independently performed in tissues from five CRS patients and five control subjects, respectively. The Mann–Whitney test was used in comparison analysis. ns not significant.

dependent on specific types of MMP, e.g., the MMP9, MMP7 and MMP10.

## Discussion

This study presents an apical-out airway organoid by using a defined hybrid hydrogel system. By utilizing this system, we have revealed a mechanism by which the epithelial-derived MMPs are required for airway epithelial ciliation, and the consequent effect of MMPs is associated with ECM (like collagen) degradation which could determine the epithelial polarity (Fig. 8). The CAH gel-based hANO model could recapitulate aspects of cellular complexity, structural characteristics, and the developmental properties of native nasal epithelium.

The degradation of ECM such as collagen is important for many tissues/organs development[9], but related studies in airway epithelium formation remain inclusive. Here we show that during differentiation, the polarization of airway epithelium needs to degrade collagen, and consequently creates a lumen within the gel, which facilitates the apical-out polarity of mucociliary cell differentiation. Interestingly, a study done in human enteroids also reports that manual removal of ECM proteins can reverse the organoid polarity from basal-out to apical-out orientation[31]. In addition, recent two studies present the airway apical-out organoids can be generated by using ECM-free system[20,21]. Both our findings and the literature reports indicate that reduced or minimal ECM approaches are essential to drive apical-out polarity in airway organoid cultures.

So far there are two methods have been introduced to create apical-out orientation of airway organoids, including ECM-free and modified Matrigel systems. Some former studies have indicated a free-floating airway apical-out epithelial spheroids, in which this is similar to the recent ECM-free method[32]. Stroulios et al. and Wijesekara et al. have described an ECM-free with low-adherence microwell method to generate the organoids with apical-out pattern[20,21]. However, organoids in these ECM-free systems present a ciliated cell predominance phenotype, and fail to differentiate goblet cells and also don't contain

basal cells or club cells; and no ECM around the organoids also implies that such system could not mimic the physiological environment in tissues. Boekig et al. has reported a two-phase culture system in a transwell insert (air-gel interface), that is, expansion and differentiation culture[33]. The epithelial cell units proliferate in mixed matrix gel components (2:1 mixture of collagen and Matrigel) for 14 days; then cell units are released by gel digestion and re-suspended in collagen for differentiation culture (another 28 days), and the organoids develop to a polarized epithelium with outwardly facing apical membranes; while the replating manipulation causes a substantial loss of organoid units (about 60%). Moreover, this model also lacks differentiated goblet cells. Goblet cell is one major functional cell type in airway epithelium, and mucus secretion is a critical response of epithelium to pathogens or environmental factors[34,35]. The absence of secretory cell lineage suggests that these apical-out organoid models could not fully represent the cellular composition of native epithelium, so that it would limit their application in studying airway epithelial development (especially those related to goblet cell differentiation) and also the application in elucidating the mechanisms of epithelial mucus production during pathogen-host interaction.

The current hANOs can mimic certain morphological and structural characteristics of native airway epithelium from proliferation to differentiation. The organoids exhibit branching morphogenesis in 3D in vitro environment during the proliferation stage, which may represent the morphological patterns of epithelial cell growth and migration in vivo tissues. One interesting phenomenon found in this organoid model is that following differentiation ECM is gradually degraded, and organoids can move and merge into larger spheroid-like structures. The recent developmental studies done in chick lungs have reported that the continuous network of airway is generated by large-scale epithelial fusion events[36,37]. During this fusion process, epithelial basal cells can contact each other at the fusion site, while the apical polarity is maintained (i.e., those differentiating or differentiated cells still locate at the apical layers); meanwhile, ECM around

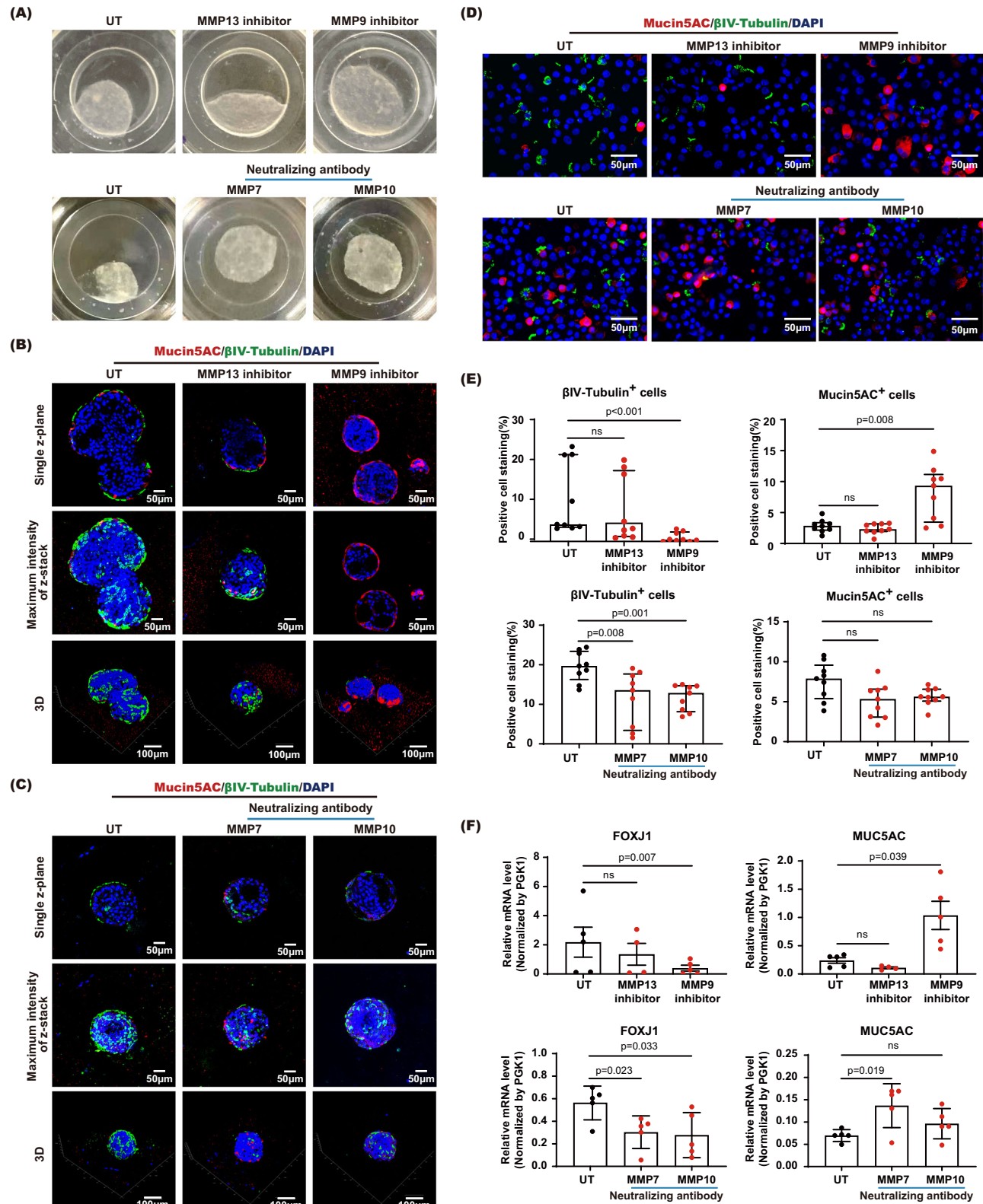

**Fig. 7 | Characterization of hANOs treated with MM9 or MMP13 specific inhibitor, and neutralizing antibody of MMP7 or MMP10.** **A**–**D** Representative pictures of gel degradation and candidate cellular markers (βIV-tubulin and MUC5AC) of hANO (by multiphoton microscopy and immunofluorescent microscopy) in differentiated hANOs (D24) treated with MMP9 or MMP13 specific inhibitor, and MMP7 or MMP10 neutralizing antibody versus untreated organoids. **E**, **F** βIV-tubulin+ cells, MUC5AC+ cells, and mRNA expression of *FOXJ1* and *MUC5AC* were compared in hANOs treated with MMP9 or MMP13 specific inhibitor, and MMP7 or MMP10 neutralizing antibody versus untreated hANOs, for positive cell analysis, data present the median with interquartile, *n* = 9 experiments per condition; for qPCR analysis, data present the mean with SD, *n* = 5 experiments per condition. The experiment was independently performed in three organoid lines from three different donors. The Mann–Whitney test was performed in (**E**); the unpaired *t* test was used in comparison analysis in (**F**). ns not significant.

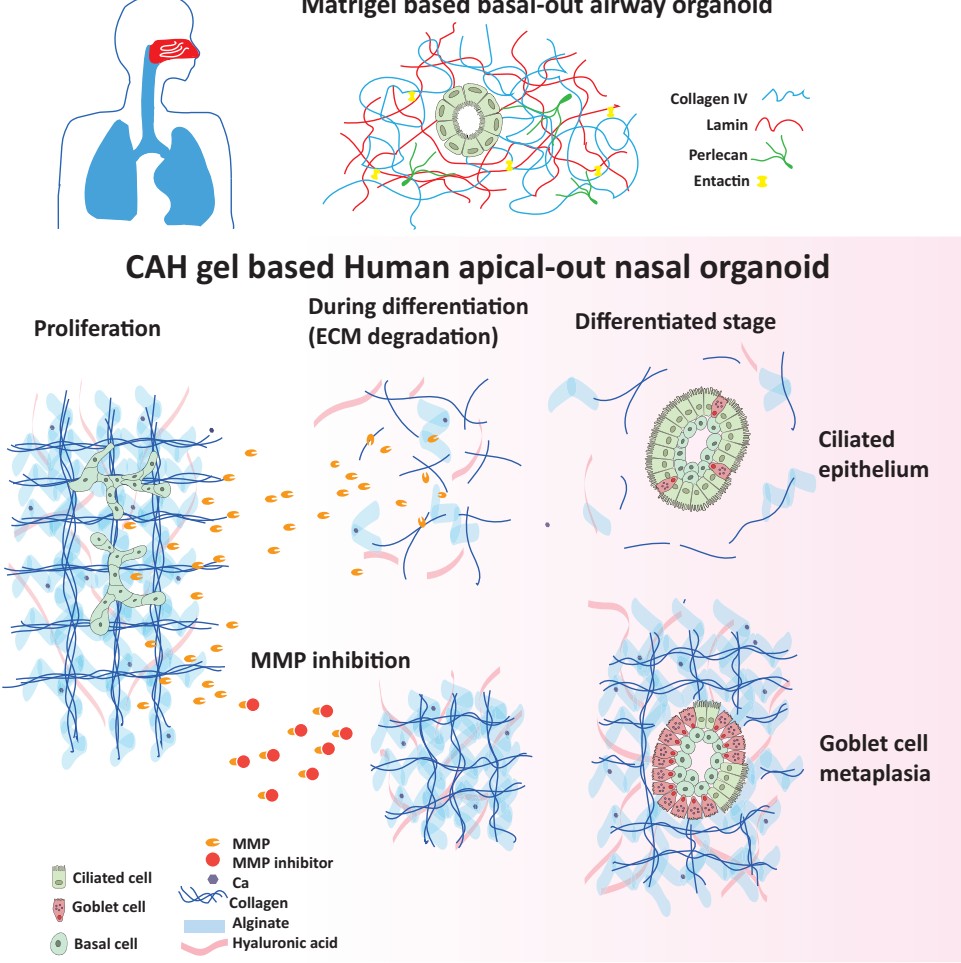

**Fig. 8 | Diagram of MMP mechanism on ECM degradation associated with epithelial ciliation and polarity by using current hANO system.** Schematic diagram shows the comparison between CAH gel based hANO and Matrigel based airway organoids, and the roles of MMPs in airway organoid development.

the fusion partners is degraded; finally, the merged epithelial structures form a larger scale epithelium. The live videos and multiphoton microscopy images in the current model also suggest that basal-to-basal docking probably mediates the organoid fusion process, similar to the epithelial fusion during lung development in vivo. Therefore, the hANO model could be a useful in vitro platform to investigate the cellular mechanism as well as their interaction with ECM during epithelial fusion in airway development.

In this hANO model, human primary nasal epithelial progenitor cells are incorporated into CAH gel. Airway epithelial progenitor cells are considered to retain intrinsic growth and differentiation characteristics of respiratory epithelial tissues from which they are derived, and alteration of epithelial inherent properties could be associated with the defects in epithelial repair or remodeling in inflamed nasal mucosa[24,38,39]. In this study, nasal epithelial cells from the middle turbinate using a minimally invasive brushing technique are compatible with CAH gel and culture condition; and the isolated cells (about $5 \times 10^4$) can grow and differentiate into matured nasal organoids without expansion by serial passages. Thus, the organoids by using epithelial cells directly derived from nasal mucosa by resected tissues or brushing method may maintain the phenotypic properties of the original cells from individual donors. The above characteristics of hANOs offer potential advantages in that hANOs could be a useful platform to study the epithelial development in healthy and disease condition, and test individual local responses to pathogens and drugs, hence facilitating precision medicine purpose.

hANOs have an outward-facing apical surface with ciliated cells, which should easily contact various exogenous factors such as bacteria, viruses, and environmental elements that could be added to the culture media. Infection with pathogens in base-out airway organoids requires a technically challenging method. One needs to extract organoids from the Matrigel, mechanically shear the cultures to allow viral access to the apical surface of the organoids, and then re-embed the organoids in the Matrigel[14,19]. Furthermore, in apical-out system the secretion factors are released into the culture media, where they can be easily removed or collected for further analysis. Therefore, hANOs are expected to be a more accessible model to study host-pathogen interactions and to discern mechanisms of the pathogenesis in airway epithelium.

Various biomaterials can induce many different cellular behaviors; therefore, the selection of a certain biomaterial should depend on the goal of research. There are three reasons why we chose type I collagen instead of type IV. Firstly, type I collagen is well-known as a pH sensitive protein that can start gelation under body temperature at neutral pH condition, while type IV collagen cannot undergo self-gelation and has been rarely used in 3D cell culture but commonly applied as a coating for cell attachment[40,41]. Secondly, although it is known that the basal lamina of respiratory epithelium is mainly composed of type IV collagen, type I collagen is still the major type of ECM component around the epithelium in the respiratory tissue which has significant impact on epithelial development[9,42]. More importantly, type I collagen can facilitate cell migration and can be degraded by MMPs[9], which thus

promote the outward growth of organoid, leading to the apical-out orientation. Hence, we choose type I collagen as the major component of this hydrogel which can better mimic the ECM microenvironment of airway epithelial tissues.

MMPs play important roles in tissue remodeling, wound healing, and morphogenesis in respiratory mucosa during both physiological and inflammatory conditions[6,43–45]. The previous studies have mostly focused on the effects of fibroblast or inflammatory cell-derived MMPs on airway inflammation[46–49], and epithelial-derived MMPs have only reported in some stimulant-induced cell culture experiments[50–52]. Nonetheless, the effects and the underlying mechanisms of MMPs on airway epithelial development and abnormal remodeling remain poorly understood. Using the hANO model as well as two conventional 3D culture systems (ALI and Matrigel-based apical-in organoids), our findings reveal that following differentiation, expression of various MMP genes (MMP7, MMP9, MMP10, and MMP13) is up-regulated in nasal epithelial cells, and the normal ciliation process is suppressed by inhibition of MMP activity, suggesting MMP is functionally required for nasal epithelial differentiation. Interestingly, when inhibiting MMP activity in hANO system, the airway epithelial cells display abnormal differentiation phenotype, i.e., increase the relative number of goblet cells. The mechanism behind MMP inhibition and its association with goblet cell differentiation is not investigated in this study; while, the RNA-seq data suppose certain molecular candidates (e.g., IL-13 and IL-1β) are up-regulated in actinonin treated organoids, in which these genes are known factors to induce goblet cell differentiation. As the interplay between epithelial cells and the surrounding ECM has essential roles in determining tissue polarity, the current hANO model is an ideal system to study the functional interaction between tissue microenvironment factors (like ECM enzymes) and ECM organization (such as collagen) on airway epithelial biology.

Our results suggest that the roles of individual MMP family members on airway epithelia development are different. For instance, MMP7, MMP9, and MMP10 have significant effects on ciliogenesis of airway epithelial cells, but MMP13 has a limited impact on it; while only MMP9 inhibition would lead to increase of goblet cell proportion. Since this study does not aim to screen the functional effects of specific metalloproteinases on epithelial development, we could not exclude the potential effects of other MMP members as well as other extracellular proteinases (e.g., A Disintegrin and Metalloproteinase with Thrombospondin motifs (ADAMTs)) on ECM, morphogenesis and development of epithelial tissues. Nonetheless, additional studies are required to validate the mechanisms of specific MMPs on regulating ECM degradation and then impacting airway epithelial development, by both in vitro and in vivo functional experiments such as conditional knockout mouse for epithelial specific MMP (e.g., MMP9).

In sum, the hANO system described here contains the biodegradable CAH gel combined with primary nasal epithelial progenitor cells isolated from human subjects, which enables the recapitulation and, therefore, advancement of the understanding of airway epithelial development. We employed hANO model to reveal the essential roles that epithelia-derived MMPs and collagen degradation process play important roles in airway epithelial cell fate determination and tissue polarity. We propose that the hANO platform has the potential to facilitate future discoveries that would be relevant for a better understanding of the biology and pathology of human respiratory tract, e.g., epithelial development, disease modeling, epithelial-pathogen interaction, and drug screening.

## Methods
### Human nasal tissues
Nasal samples (from sinus mucosa) were obtained from patients (n = 23, 16 male/7 female, median age 43) with chronic rhinosinusitis (CRS) who underwent functional endoscopic sinus surgery in the Department of Otolaryngology, The First Affiliated Hospital, Sun Yat-

sen University (Guangzhou, China). Nasal biopsies (from middle turbinate) obtained from subjects (n = 10, 7 males/3 females, median age 36) with nasal septal deviation or nasal fracture that underwent septoplasty surgery served as control mucosa. None of the CRS patients and controls had immunodeficiencies, autoimmune diseases, tumor, or acute respiratory tract infection 1 month before inclusion. To confirm the subjects without infection was based on patient symptoms report (including fever, cough, runny nose, sore throat, chest or nasal congestion, and fatigue) and laboratory tests (nasal swabs sent for bacterial culture test and nucleic acid amplification tests for SARS-COV-2). All subjects were carefully selected and only individuals who did not receive systemic glucocorticoid steroids, antibiotics for 3 months and/or nasal corticosteroids for 1 month before the surgery. Approval for this study was obtained from the Institutional Review Board of The First Affiliated Hospital, Sun Yat-sen University (Project number [2017] 256). All participants were given written informed consent. Relevant approvals from China's Ministry of Science and Technology related to the export of genetic information and materials relevant to this work was obtained.

### Human nasal epithelial progenitor cell (hNEPCs) cultures
Nasal tissues from CRS patients were cut into pieces and processed by enzymatic digestion with Dispase II (Sigma-Aldrich, St.Louis, MO) at 4 °C overnight. The digested tissue suspension was washed by cold Dulbecco's Modified Eagle Medium (DMEM) (Thermo Fisher Scientific, Waltham, MA) and filtered by using 70 μm cell strainers (CORNING, Corning, NY) to obtain single-cell suspension. The human nasal epithelial progenitor cell (hNEPC) culture method was modified based on our previous reports[24,53]. Briefly, primary cells were seeded on the X-ray treated 3T3 cell feeder and were cultured with in-house serum based growth medium (including DMEM/F12, fetus bovine serum, cholera toxin, insulin, EGF, hydrocortisone, 3,3',5-triiodo-L-thyronine, adenine, and ROCK inhibitor); only progenitor cells can grow clonally and were collected after 80% confluence for organoid culture.

### hANO cultivation
CAH hydrogel was prepared by modifying the previous protocol[22,23] with addition of hyaluronic acid. hNEPCs were re-suspended in cold premixed gel solution (1 ml size) which included collagen type I (Ibidi GmbH, Grafelfing, Germany), low viscosity alginic acid sodium salt stock solution (Sigma-Aldrich), hyaluronic acid (Sigma-Aldrich), 10x DMEM (60 μl) (Thermo Fisher Scientific), and growth medium (250 μl). The final concentrations of gel components were 3 mg/ml collagen, 5 mg/ml alginate, and 1 mg/ml hyaluronic acid. The premixed solution (250 μl per dish) with hNEPC suspension was adjusted to neutral pH, cross-linked with 3.75 mM CaCl₂ (Sigma-Aldrich) and then was gently placed in a 15 mm glass bottom cell culture dish (Nest Biotechnology, Wuxi, China). For each culture dish, the number of seeding hNPECs was $1 \times 10^4$. The above procedures were carried on ice. The premixed gel solution-hNEPC suspension was incubated for 30 min at 37 °C to solidify the CAH gel; then growth media (500 μl) were covered the gel and the cultures were incubated at 37 °C with 5% $CO_2$. During the proliferation stage (Day 1 to Day 10) of hANOs, growth medium was replaced every 2 days; during the differentiation stage (Day 11 to Day 24) of hANOs, differentiation medium (500 μl) (PneumaCult™-ALI Medium, STEMCELL Technologies, Vancouver, Canada) was changed every 2 days. The images of organoids were captured by phase-contrast microscopy using a 10x or 20 x objective on IX51 inverted microscope (Olympus, Tokyo, Japan). The real-time organoid behavior was monitored by Lux3 FL live-cell imaging microscope (CytoSMART Technologies B.V., Eindhoven, The Netherlands). Time-lapse videos were generated to display the structural changes of organoids over proliferation and differentiation. The beating cilia of hANOs were observed and captured by using the Sisson-Ammons Video Analysis system (SAVA, Omaha, NE)[54].

### Immunofluorescent (IF) staining and multiphoton microscopy

The primary antibodies for specific cell types were used to characterize hANOs: mouse anti-human βIV-Tubulin monoclonal antibody [clone ONS.1A6], rabbit anti-human ZO-1 polyclonal antibody, rabbit anti-human Mucin 5AC monoclonal antibody [clone EPR16904], mouse anti-human p63 monoclonal antibody [clone 4A4], and rabbit anti-human Foxj1 monoclonal antibody [clone EPR21874]; all antibodies were purchased from Abcam (Abcam, Cambridge, UK). The collagen hybridizing peptide 5-FAM conjugate (F-CHP) was synthesized and provided by Prof. Yang Li from the Fifth Affiliated Hospital of Sun Yat Sen University (Zhuhai, China)[29].

The organoids with CAH gel were directly fixed by 4% paraformaldehyde for 30 min and were permeabilized with 0.3%100x-triton for 15 min. The organoids were incubated by primary antibodies at 4 °C overnight (the dilutions of antibodies targeted to p63, βIV-Tubulin, Mucin 5AC, foxj1, and ZO-1 were 1:100, 1:800, 1:600, 1:1000, and 1:500, respectively), and then stained with goat anti-mouse Alexa Fluor 488 and goat anti-rabbit Alexa Fluor 594 conjugated secondary antibodies (Thermo Fisher Scientific) in the dark at room temperature for 1 h; the samples were then stained with 4′,6′- diamidino-2-phenylindole (DAPI) for visualization of the nuclei; finally the organoids with CAH gel were covered by PBS and stored at 4 °C in dark. With regard to the F-CHP staining, the collagen conjugates were diluted by PBS and heated at 80 °C for 5 min followed by cool-down on ice. The pretreated F-CHP (5 μM) was incubated in organoids with CAH gel at 4 °C overnight. Images of the above IF staining in hANOs were captured by using 25x water immersion objective on a TCS SP8 DIVE multiphoton microscope (Leica, Mannheim, Germany). Images were 3D-reconstructed using LAS X software (Leica). For analyzing the degree of collagen degradation, the ratios of F-CHP positive signals to the total hydrogel signals in three randomly selected fields were calculated by using Imaris 8.4 software (Bitplane, Zürich, Switzerland).

### Cytospin experiment and cell counting

CAH gel was first degraded by sodium citrate (50 mg/ml) and type I collagenase (1 mg/ml), and organoids were digested by Accutase (Thermo Fisher Scientific) to obtain single cell suspension. About $1 \times 10^4$ resuspended cells in 200 μl of PBS were loaded in the cytospin chamber; cell suspension was centrifuged at 500 g for 5 min by using Cytospin®4 Cytocentrifuge (Thermo Fisher Scientific); and then the cells were collected on glass slides for subsequent IF staining experiment which was described in the above section. Images of samples stained with specific markers for ciliated cells (βIV-Tubulin) and goblet cells (Mucin 5AC) were captured at 200x or 400x magnification under DM4B upright microscope (Leica). The βIV-Tubulin⁺ or Mucin 5AC⁺ cells were counted in three randomly selected areas (200x magnification) and the percentages of ciliated cells and goblets cells among total epithelial cells (more than 300 cells totally) were calculated.

### RNA-seq experiment and data analysis

RNA from hANOs on Days 10, 17, and 24 was isolated by using Trizol reagent (Thermo Fisher Scientific). RNA quality was checked by Bioanalyzer 2100 system (Agilent Technologies, Santa Clara, CA). The library preparation was carried out following the manufacturer's protocol by using NEBNext® UltraTM RNA Library Prep Kit for Illumina® (NEW ENGLAND BioLabs, Ipswich, MA). cDNA libraries were sequenced on an Illumina Novaseq platform as 150-bp pair ended reads (illumina, San Diego, CA).

Kallisto program was used to quantify the abundances of transcripts from RNA-seq data[55], and the Kallisto index was built with reference transcriptome GRCh38. Reads per kilobase per million mapped fragments (RPKM) counts and differential expression of gene transcripts were estimated using DEseq2 program (Bioconductor, https://www.bioconductor.org). The difference of transcriptomic data were analyzed by two-sided Wald test methods and the multiple

comparison correction were analyzed by Benjamini–Hochberg methods. Differentially expressed genes (DEGs) were selected based on the cut-off at a P value of less than 0.05 and the absolute value of Log2 fold change greater than or equal to 1.5. For intra-group analysis, i.e., different differentiation days of hANOs, DEGs were analyzed in samples from Day 10, Day 17, and Day24 at pairwise comparison. For inter-group analysis, i.e., hANOs treated with MMP inhibitor versus untreated hANOs, DEGs were analyzed between MMP inhibitor treated and untreated samples on Day 17 or Day 24. ClusterProfiler program[56] was used to perform enrichment analysis, including Gene Ontology (GO) term analysis, KEGG pathway analysis and GSEA analysis based on the DEGs from above intra-group or inter-group comparisons; the above functional analyses were analyzed by two-sided Over Representation Analysis (ORA) with Benjamini–Hochberg multiple comparison correction.

### Quantitative RT-PCR

Total RNA was reversed transcribed to cDNA by using PrimeScript RT kit (Takara, Shiga, Japan) and was subsequently analyzed by quantitative RT-PCR using the FastStart Universal SYBR Green Master (Roche, Basel, Switzerland). The expression levels of target genes were normalized to housekeeping gene PGK1. Primer sequences for MMP genes were:

1) *MMP7*
Sense: GAGTGAGCTACAGTGGGAACA;
Antisense: CTATGACGCGGGAGTTTAACAT;
2) MMP9
Sense: AGACCTGGGCAGATTCCAAAC;
Antisense: CGGCAAGTCTTCCGAGTAGT;
3) MMP10
Sense: TGCTCTGCCTATCCTCTGAGT;
Antisense: TCACATCCTTTTCGAGGTTGTAG;
4) MMP13
Sense: TCCTGATGTGGGTGAATACAAT;
Antisense: GCCATCGTGAAGTCTGGTAAAAT.

### Luminex assays

MMP protein levels (MMP7, MMP9, MMP10, and MMP13) in secretion of hANO cultures were measured by human Magnetic Luminex Performance Assay MMP base kit (R&D systems, Minneapolis, MN) following the manufacturer's instructions. The experiment was performed by using a Luminex MAGPIX system (Luminex, Austin, TX). The protein expression levels were measured as mean fluorescence intensity (MFI) and the concentration of each MMP protein was calculated using a 5-parameter logistic fit curve generated for each analyte from the standards. The detection sensitivity for each selected MMP was 6.6 pg/ml (MMP7), 13.7 pg/ml (MMP9), 3.2 pg/ml (MMP10), and 63.5 pg/ml (MMP13) respectively.

### Measurement of MMP activity

The general MMP enzyme activity in hANO supernatants was detected by using MMP Activity Assay Kit (Abcam), which included a fluorescence resonance energy transfer (FRET) peptide as a generic MMP activity indicator. Recombinant human MMP9 protein (BioLegend, San Diego, CA) served as positive control, was pre-activated with 1 mM of 4-Aminophenylmercuric Acetate (APMA) for 24 h at 37 °C. The hANO supernatants, APMA pre-treated MMP9, and negative control (only culture medium) were added in duplicate into a 96-well plate; MMP Green substrate was applied into the assay plate; and fluorescence units were measured using a VICTOR Nivo Multimode Plate Reader (PerkinElmer, Waltham, MA) at Ex/Em = 490/525 nm.

### MMP inhibitory assay in hANOs

The inhibitor for multiple MMP proteases (Actinonin) (Chemical formula $C_{19}H_{35}N_3O_5$, Enzo, Farmingdale, NY), MMP9 specific inhibitor

(Chemical formula $C_{16}H_{17}F_2N_3O_3S$, Sigma-Aldrich), and MMP13 specific inhibitor (Chemical formula $C_{22}H_{20}F_2N_4O_2$, Sigma-Aldrich) were used to perform MMP inhibition assay. Actinonin can inhibit the activity of MMP1, MMP2, MMP3, MMP7, MMP8, MMP9, MMP10, MMP12, and MMP13. hANOs were treated with Actinonin (25 µM), MMP9 inhibitor (100 µM), and MMP13 inhibitor (10 µM) twice a day, respectively, from Day 10 (when the organoids started differentiation) to Day 24. MMP activity was measured in the supernatant of hANOs collected on Day 17 and Day 24. The dosages of actinonin, MMP9 specific inhibitor and MMP13 specific inhibitor were modified based on the manufacturer's manual and published articles[57–59].

### Masson staining and immunohistochemistry experiment
Nasal tissues were obtained from 6 patients with nasal polyps and 6 control subjects. Samples were fixed with 4% paraformaldehyde and embedded in paraffin. Tissue sections were performed Masson Trichrome staining to analyze the thickness of basement membrane beneath epithelium; while immunohistochemistry (IHC) was used to evaluate MMP protein expression patterns in normal epithelium and epithelium with goblet cell hyperplasia. Antibodies used in IHC experiment were as follows: rabbit anti-human MMP7 monoclonal antibody [clone EPR17888-71] (Abcam), rabbit anti-human MMP9 monoclonal antibody [clone EP1254] (Abcam), mouse anti-human MMP10 monoclonal antibody [clone 110304] (R&D systems), and rabbit anti-human MMP13 monoclonal antibody [clone EPR21778] (Abcam).

For IHC staining procedures, tissue sections were first performed antigen retrieval step with target retrieval buffer (pH6.0) followed by incubation with respective primary antibodies for MMP7, MMP9, MMP10, and MMP13 at 4 °C overnight; and then the sections were incubated with Dako REALTM EnVisionTM Detection System (Agilent, Santa Clara, CA) and stained with diaminobenzidine substrates for color development; all slides were counterstained with hematoxylin.

The evaluation of basement membrane and MMP proteins were performed in three randomly selected regions at a 200x magnification on DM4B upright microscope (Leica). The thickness of basement membrane was measured by using Image J software. Expression levels of individual MMP protein was evaluated based on extensity (positive staining areas of total epithelial area) and intensity (average optical density) by using Image J software.

### Statistics and reproducibility
Unpaired or paired *t* test and Mann–Whitney *U* test was used to analyze the significance between two groups based on the data distribution, respectively; Friedman test followed by Dunn's multiple comparison tests was used to analyze the MMP expression data among three groups; while Two-way ANOVA test followed by Tukey's multiple comparison tests was used to analyze the MMP activity data. All statistical analyses and graphs were performed by using GraphPad Prism version 8.0 (GraphPad Software, La Jolla, CA). The *p* value is calculated for a two-tailed test; $p < 0.05$ was considered as statistically significant.

The numbers of images taken for organoids at different conditions are described as bellows.

In Fig. 1A, 5 micrographs of bright field microscopy were captured for each time point; 5 multiphotonic graphs of p63 staining were captured for each time point; 4, 8, and 10 multiphotonic graphs of βIV-tubulin and MUC5AC staining were captured for D10, 17 and D24, respectively. In Figs. 1B, 3 multiphotonic graphs of FOXJ1 and ZO-1 staining were captured for D24. In Fig. 1C, 3 gel degradation graphs were captured for each time point; 3 micrographs of F-CHP staining were captured for each time point.

In Fig. 4B, 3 multiphotonic graphs of βIV-tubulin and MUC5AC staining were captured for untreated (NT) and actinonin treated organoids, respectively. In Fig. 4E, 3 gel degradation graphs were captured for untreated (NT) and actinonin treated organoids, respectively. In Fig. 4F, 6 micrographs of F-CHP staining were captured for each time point in organoids treated with and without actinonin, respectively.

In Fig. 7A, 3 gel degradation graphs were captured for untreated (NT) and specific MMP inhibitor treated organoids, respectively. In Fig. 7B, C, 2 multiphotonic graphs of βIV-tubulin and MUC5AC staining were captured for organoids treated with and without specific MMP inhibitors, respectively. In Fig. 7D, 3 micrographs of βIV-tubulin and MUC5AC staining were captured for organoids treated with and without specific MMP inhibitors, respectively.

### Reporting summary
Further information on research design is available in the Nature Portfolio Reporting Summary linked to this article.

## Data availability
RNA-seq data in this study have been deposited in the Gene Expression Omnibus (GEO) under series GSE249160 at https://www.ncbi.nlm.nih.gov/geo/query/acc.cgi?acc=GSE249160. The data supporting the findings of this study are available within the article and its supplementary materials. Source data are provided with this paper. The authors declare that there are no restrictions on the data availability of the results presented within this study and all material is available upon request from the corresponding authors. Source data are provided with this paper.

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

## Acknowledgements

The authors thank Prof. Marius Sudol for editing the manuscript; and thank Prof. Yang Li from the Fifth Affiliated Hospital of Sun Yat Sen University for providing the collagen hybridizing peptide 5-FAM conjugate (F-CHP). This study was supported by grants from the National Natural Science Foundation of China 81974139, 82171112 and 81770983 to C.Li, 32101062 to C.Liu, 82201261 to L.Li, and 81974141 to J.Li; Open Project of State Key Laboratory of Respiratory Disease SKLRD-OP-202008 to R.Chen and C.Li; Guangzhou Science and Technology Program 201907010038 to C.Li; and Guangdong Basic and Applied Basic Research Foundation 2022A1515012607 to C.Liu.

## Author contributions

L.L.: conceptualization, experimental operation, investigation, formal analysis, data curation, writing – original draft; L.J.: experimental operation, investigation, formal analysis, data curation, writing – original draft; D.F.: RNAseq data analysis, methodology, formal analysis, data curation; Y.Y.: image creation, RNAseq data analysis, data curation; X.Yang: image creation and analysis, data curation; J.L.: patient recruitment, project administration, funding acquisition; D.J.: experimental operation, image creation, formal analysis; H.C.: patient recruitment, sample collection and procession, project administration; Q.M.: sample collection and procession, project administration; R.C.: sample collection, fund acquisition; B.F.: experimental operation, methodology; X.Z.: project design, methodology; Z.L.: project design, RNAseq data analysis; X.Ye: experimental operation; Y.H.: project design; C.Liu: project administration, project design, funding acquisition, writing – review & editing, supervision, project administration. C.Li: project administration, project design, funding acquisition, writing – original draft, writing – review & editing, supervision, project administration.

## Competing interests

The authors declare no competing interests.
