## [Peer Review File · Nature Communications]

Human apical-out nasal organoids reveal an essential role of matrix metalloproteinases in airway epithelial differentiationEditorial Note: Parts of this Peer Review File have been redacted as indicated to remove third-party material where no permission to publish could be obtained.

REVIEWER COMMENTS

Reviewer #1 (Remarks to the Author):

- What are the noteworthy results?

This manuscript shows a novel and innovative approach to organoid culture that overcomes some of the key disadvantages to airway organoid culture, which limits utility for a variety of applications. The authors further demonstrate interesting and novel findings regarding the interaction of an artificial extracellular matrix that highly recapitulates that in vivo, and identifies novel roles for MMPs in both the interaction with ECM as well as ciliogenesis.

- Will the work be of significance to the field and related fields? How does it compare to the established literature? If the work is not original, please provide relevant references.

The manuscript is of general significance to the study of epithelial biology and presents a new method of study which can be applied to several disease areas. It has several novel features but lacks some appropriate reference to prior work which are easily identified. Overall, however, the work is quite novel and does not replicate prior work.

- Does the work support the conclusions and claims, or is additional evidence needed?

Overall the work adequately supports the conclusions and claims with a few minor clarifications needed.

- Are there any flaws in the data analysis, interpretation and conclusions? - Do these prohibit publication or require revision?

There are some flaws in the analysis as described below requiring revision.

- Is the methodology sound? Does the work meet the expected standards in your field?

The methodology is sound. Both well-established and novel techniques are used in a complementary manner and the work meets expected standards.

- Is there enough detail provided in the methods for the work to be reproduced?

Overall, there is sufficient detail provided to replicate the work, except as noted below.

Specific critiques.

1) Only diseased nasal mucosa was sampled for the main study cultures according to results, using endoscopy. This may not be accurate because methods indicates controls. However, there are no clear distinction on controls versus CRS in the manuscript for cultures. This must be clearly delineated and clarified in the results, methods, and all figures.

2) There are numerous manuscripts describing minimally invasive collection of nasal epithelial cells using brushes or curettes, which would enable similar numbers of basal progenitor cells to be collected, based on the plethora of manuscript employing a similar method. The authors do not describe why they opted only for diseased tissue available through endoscopy, which limits applicability to other populations and study design. It calls into question whether invasive nasal tissue is required to replicate the results found in this study, or if less invasive means may be used.

- 3) The authors state that in later stages of cultures, the organoids' branched structures appear to coalesce and merge into larger, spheroid-like structures that are not hollow. Clarify how this process is consistent with in vivo epithelial airway generation.
- 4) The authors state that in Fig 1A clearly demonstrated is the p63 predominance of cell type. However, the image provided does not represent any green p63 staining. Recommend selecting an alternative. Also, the authors do not describe evaluating comprehensively for airway epithelial types (including non-ciliated non-goblet cells, such as ionocytes). The ZO-1 staining in the provided image is blurry and typical epithelial morphology is not apparent. This is not convincing of tight junction staining. Further the authors state that the pattern of beta-tubulin staining suggests apical out ciliation. Videos do support this statement but the image in figure 1 is not of sufficient resolution to be conclusive.
- 5) Authors state "degraded collagen may produce bioactive fragments that likely promote organoid differentiation" – this statement is unclear and not supported by results, should be removed or supported by additional data.
- 6) For Figure 1, only the collagen experiments are clearly labeled with sample size. Please clarify the sample size for all imaging experiments completed in this figure and all figures. Furthermore, the authors state sample size as # of experiments in the captions, but do not clarify if these are solely experimental replicates or if there are technical replicates also for each experiment.
- 7) Some of the videos have organoid structures that appear to have something inside them. It could be that some are apical-in structures or artifacts of the time-lapse. This should be clarified in a caption or description for the video.
- 8) Many of the figures reproduce with such small font as to be illegible. All figures must be legible by the average viewer. Even the higher resolution figures had to be increased in size significantly to be legible.
- 9) The authors do not state if the gene expression of airway epithelia differs in this model compared to apical-in organoids or monolayer cultures, and whether this is or is not consistent with epithelial differentiation in vivo (for example, in an animal model). This is also a critique for later experiments in the organoid models. Are there other experiments in other model types that could lend additional credence that these findings are not unique to the specific hydrogel or experimental conditions? The findings could be at least somewhat replicated in other culture systems, including monolayers, and the authors should comment on whether there are any other studies in other model types that may also be supportive of the findings in this manuscript.
- 10) The authors state that they selected specific MMPs for further evaluation. Please state why the MMPs selected were chosen – current wording in the text suggests that there are other MMPs that may have been altered. The authors should clarify if there are any other known proteins that participate in ECM degradation or remodeling and if they also evaluated those genes for altered expression during differentiation as well, from RNA-seq data. This would be more comprehensive and inclusive of possible contributors to the ECM degradation, although further evaluation beyond the RNA-seq would not be considered within scope of this manuscript.
- 11) Video S5 is suggested to show no evidence of cilia, however, the video is zoomed in too far to see much of the apical surface. This is not conclusive of no cilia.
- 12) In the discussion the authors report only 3 studies of apical out airway organoids. However, in fact, there are many going back to the 1990s. However, it may be correct that there are a limited number of culture apical out airway organoids in an artificial extracellular matrix. The authors should revise this sentence to reflect accurately the publications of this type of model.
- 13) The authors must add references on page 19 to the discussion where they are describing

characteristics of different collagens.

14) Methodology: authors state that patients did not have infection in the month prior to sampling. However, they do not describe testing. Was this conclusion based on patient report or some viral/bacterial testing?

Reviewer #2 (Remarks to the Author):

Li et al's manuscript, titled "Primary human apical-out nasal organoids reveal an essential role of matrix metalloproteinases in airway epithelial differentiation and remodelling," describes a novel culture method for apical-out oriented three-dimensional spheroids/organoids of the upper airway that can differentiate into all known major cell types of the mucociliary tract. The authors demonstrate that perturbation of matrix metalloproteinases (MMPs) can alter cellular differentiation.

However, while the story is well written and concise, the manuscript may have limited impact on the field of airway organoids in its current form. Specifically, the manuscript lacks fundamental insights into the presented mechanism of MMPs and cellular differentiation, and the translational value of the results is not well explored or explained, thereby limiting its impact. In conclusion, the manuscript appears to be more technology-based than biology-based. Therefore, it is important for the authors to provide a more detailed discussion on the potential applications of this technology in order to demonstrate its relevance and impact on the field of airway organoids.

Major points

1. The authors introduce a new hydrogel-based technology for generating apical-out organoids. However, a previous method for generating apical-out organoids has been reported by Stroulios et al. While the authors briefly discuss the comparison between the two methods, a direct comparison of their efficiency in differentiation would greatly benefit the manuscript.
2. The authors utilized human nasal epithelial progenitor cells (hNEPCs) to generate the spheroids. However, it is unclear whether each of the 17 patients was used individually or if a pooled fraction was used, which raises concerns about the efficiency and robustness of the culture method. The manuscript lacks statistics on the number of lines generated from the 17 donors, and as a result, the findings may be interpreted as being based on a single hANO line. To identify common mechanisms and robustness of the system, the results should be repeated with different lines from different donors.
3. The authors emphasize the significance of apical-out organoids in the mechanisms described, including the differentiation toward ciliated cells and the shift toward goblet cell differentiation upon MMP inhibition. To demonstrate the importance of apical-out organoids, the authors could compare their system with conventional apical-in organoids in Matrigel to highlight the added value of their approach. The absence of such a comparison makes it challenging to assess the value of the presented system.
4. The authors state that the spheroids represent an epithelium that mimics the in vivo situation. However, in line 161, the authors suggest that the spheroids have no lumen and are apical-out oriented. This contradicts Figure 1A (MUC5AC d24) and 1B, which shows that the organoids have small lumens containing polarized cells, as indicated by the ZO-1 staining. This suggests that the epithelium is double-

polarized, with both apical-out and apical-in orientations. It is debatable whether such an epithelium is representative of the airway epithelium. Additionally, the authors do not address the question of whether the presence of collagen I on the apical side of cells is representative of the in vivo epithelium, which normally has its apical surface oriented toward the air. This issue is further highlighted by the patient data in Figure 6, which shows differences in basement membrane thickness that is basally located and therefore opposite to the apical-out method presented in this study.

5. The mechanism presented is currently only correlated to disease and goblet cell differentiation. The manuscript would greatly benefit genetic ablation of specific MMPs to better identify the biological relevance. Moreover, it would greatly benefit the impact of the new culture system.

6. In their supplementary video S2 and S3 there are clearly organoids moving through the viewing field. The authors do not comment on this or explain the significance.

Minor points

1. In line 233, the authors state that they examined all MMP genes, but they do not provide this data. Including this information in the manuscript would allow the authors to provide stronger evidence for the biological significance of the MMPs they eventually highlight. Additionally, it would be advantageous to investigate this in various lines from different donors (see major comment 2)

2. Figure 1D shows that the highest MMP activity is observed in day 10 organoids, which consist of proliferating basal cells. This result is unexpected, yet the authors do not offer any commentary or explanation on it.

3. Line 255-258 claims differences in morphology in the organoids. While the brightfield images are convincing, the authors could strengthen their point by quantifying the circularity of the organoids.

4. Video S5 is very short and therefore not relevant to view

5. The mechanism behind MMP9 inhibition and its association with goblet cell differentiation remains unexplored in this manuscript. If the authors can provide more insight into this mechanism, it would enhance the biological relevance of their findings. For instance, is this solely due to the spatial requirements for generating cilia, or is there a signalling cascade that is altered, leading to differentiation towards an alternative pathway? There are various studies that have been conducted on the differences of ciliated and goblet cell differentiation, which the authors could examine to identify potential mechanisms.

6. All heatmaps presented have no clear legend that indicates which values are presented. The colour scale is thereby imbalanced around -0.5 instead of 0.

7. In figure 3B-C, the authors use line graphs to indicate differences in expression. This however pretends that the same culture was followed over time and time points were isolated from the exact same organoids. This is not feasible. The authors should therefore replace the line graph with bar graphs and statistics.

Textual/visual points

1. Line 49 has spelling error "to access" should be "accessing the"

2. Line 51 misses commas for easier reading

3. Line 93 misses "a" before murine tumor

4. The word system in line 94 is not needed

5. Line 99-100: "... cells in Matrigel more tend to grow inwards..." should be "... cells in Matrigel tend to grow more inwards..."

6. Line 110 has one capital too many
7. Line 111 has one space too many
8. Line 196 has spelling error in “genes”
9. Figure 1D is very small

Reviewer #3 (Remarks to the Author):

This manuscript describes the preparation of a biochemically-defined hybrid gel comprised of collagen type I, alginate, and hyaluronic acid. Human nasal epithelial progenitor cells embedded in this gel degrade the ECM to create hollow spaces within which the cells differentiate into ciliated cells. Addition of protease inhibitors suppress ECM degradation, reduce ciliation and promote mucus-secreting cell production.

Gene expression studies show increase in matrix metalloproteinases (MMP7, MMP9, MMP10 and MMP13) during differentiation into ciliated cells. Moreover, a decrease of MMPs was found in goblet cell hyperplastic epithelium obtained from patient tissue. The patient tissue histology is very nice.

While the novel material system is interesting and impacts nasal cell differentiation, how well the observations recapitulate development is not clear and not convincing since alginate is not very physiological and presence of type I collagen and hyaluronic acid but not other ECM or basement membrane proteins also make the system artificial. It may be better to simply state that the materials system is interesting for producing a type of apical out organoid without suggesting similarity to development.

The method is advantageous to some other apical-out airway organoids as there is no need to manually remove ECM. On the other hand, minimal ECM method also exist that avoid this difficulty. Easy pathogen accessibility is a potential advantage to be demonstrated. But several other apical out airway organoids reported may also have the same advantage.

Page 9 the comment “degraded collagen may produce bioactive fragments that likely promote organoid differentiation”. is a bit hollow. That is this is hard to appreciate, without some reference, analysis or other evidence to back up the comment.

Page 9 “system could morphologically and functionally recapitulate the physiological features of human upper airway epithelium.” In what way is the author suggesting the system is morphologically physiological? Isn't the morphology opposite of physiological?

Is MMP upregulation during ciliogenesis an artefact of the culture in the engineered hydrogels or a physiological reality and necessity? MMP inhibitor addition leads to a more “branching” type of morphology. But is this really branching or just attachment to the ECM of fragments in various shapes

due to just attachment to the hydrogel? More evidence of actual branching would be needed to make these claims.

Page 17 “better recapitulate the biological properties of native airway epithelial epithelium, from proliferation to differentiation” what the basis is for the authors to conclude this is not clear. What specific findings are you pointing to when you make this claim? In particular, some of the findings, such as disappearance of basal cell types near completely from the organoids suggest the cultures may not be physiological.

Page 18. Also, while the authors state ref 21,22, 25 are not physiological for not having goblet cells, the current system losing almost all of their basal cells is also not physiological.

Ref 44-46 suggest artificial environment promote MMPs. Could the situation here be just the same? The authors do show differences in MMP in goblet hyperplasia later in life in patient tissue samples but whether MMP is involved during DEVELOPMENT as state by authors is not definitive.

Several claims such as role of collagen fragments, role of MMPs are by the authors own statements still speculative.

Schematic in Fig 8 is a misrepresentation. The authors histology and text say there is hardly any basal cells (very little p63) and no lumen.

In summary, the description of apical out nasal organoids is interesting. And the hydrogel material may be somewhat novel. MMP differences in normal vs diseased patient samples is convincing and supported by beautiful histology.

The authors are inaccurate or overly speculative in their analysis, description, and depiction (for example Figure 8).

Point-by-point responses to reviewer comments

Re: NCOMMS-23-11129, Primary human apical-out nasal organoids reveal an essential role of matrix metalloproteinases in airway epithelial differentiation and remodeling

REVIEWER COMMENTS

Reviewer #1 (Remarks to the Author):

- What are the noteworthy results?

This manuscript shows a novel and innovative approach to organoid culture that overcomes some of the key disadvantages to airway organoid culture, which limits utility for a variety of applications. The authors further demonstrate interesting and novel findings regarding the interaction of an artificial extracellular matrix that highly recapitulates that in vivo, and identifies novel roles for MMPs in both the interaction with ECM as well as ciliogenesis.

- Will the work be of significance to the field and related fields? How does it compare to the established literature? If the work is not original, please provide relevant references.

The manuscript is of general significance to the study of epithelial biology and presents a new method of study which can be applied to several disease areas. It has several novel features but lacks some appropriate reference to prior work which are easily identified. Overall, however, the work is quite novel and does not replicate prior work.

- Does the work support the conclusions and claims, or is additional evidence needed? Overall the work adequately supports the conclusions and claims with a few minor clarifications needed.

- Are there any flaws in the data analysis, interpretation and conclusions? - Do these prohibit publication or require revision?

There are some flaws in the analysis as described below requiring revision.

- Is the methodology sound? Does the work meet the expected standards in your field?

The methodology is sound. Both well-established and novel techniques are used in a

complementary manner and the work meets expected standards.

- Is there enough detail provided in the methods for the work to be reproduced?

Overall, there is sufficient detail provided to replicate the work, except as noted below.

Reply: Thanks for the above comment. We have provided point-by-point response to the following specific critiques.

Specific critiques:

1) Only diseased nasal mucosa was sampled for the main study cultures according to results, using endoscopy. This may not be accurate because methods indicate controls. However, there is no clear distinction on controls versus CRS in the manuscript for cultures. This must be clearly delineated and clarified in the results, methods, and all figures.

Reply: Thanks for the comment. Since the diseased nasal mucosa is convenient to collect during the surgery, in the manuscript we used the cells isolated from middle turbinate mucosa from CRS patients to do organoid cultures. The control nasal mucosa used in the current study aims to compare the MMP expression pattern in healthy epithelium versus remodeled epithelium from CRS mucosa. We have indicated in Results (**Page 18, Line387-389**), Methods (**Page 35, Line769-771**), and Figure 6 legend (**Page 51, Line 1158-1159**). As the main goal of this study is to establish a novel apical-out airway organoid by using a defined hybrid hydrogel system, we haven't compared the patterns or phenotypes of the current organoids derived from epithelial cells from controls versus CRS patients. But our internal data showed that the epithelial cells from control subjects can grow into the same apical-out organoids in this culture system.

Nonetheless, in this revision, we have successfully performed the organoid cultures in which the epithelial cells from healthy mucosa of control subjects and mucosa of CRS patients by nasal brushing methods; and we have added such results in the revised manuscript. The details of these results are described in the following Question 2.

2) There are numerous manuscripts describing minimally invasive collection of nasal epithelial cells using brushes or currettes, which would enable similar numbers of basal progenitor cells to be collected, based on the plethora of manuscript employing a similar method. The authors do not describe why they opted only for diseased tissue available through endoscopy, which limits applicability to other populations and study design. It calls into question whether invasive nasal tissue is required to replicate the results found in this study, or if less invasive means may be used.

Reply: Thanks for the comment. We have applied nasal brushing techniques during the endoscopy examination in clinic to obtain the mucosal cells from middle meatus from control subjects and CRS patients; and then we sort the total epithelial cells (CD45-CD90-CD142+ cells) via flow cytometry. The epithelial cells (in an optimal density 5×10^4 /well) were directly cultured in the hybrid hydrogel system, and the cells could grow and differentiate without pre-2D expansion procedure. The organoids from both control and CRS subjects proliferate and differentiate in same patterns as those from tissue samples. Thus, the current organoid system can be compatible to the cells obtained by minimally invasive collection approach like brushing technique. We have added these results (**Page 10, Line 204-214**) and figures (**New Figure S4** in Supporting information) in the revised manuscript.

3) The authors state that in later stages of cultures, the organoids' branched structures appear to coalesce and merge into larger, spheroid-like structures that are not hollow. Clarify how this process is consistent with *in vivo* epithelial airway generation.

Reply: Some developmental studies done in chick lungs have shown that the continuous network of airway is generated by large-scale epithelial fusion events [**ref** PMID: 32510705; PMID: 35800889]. The basic evident of this epithelial fusion is that two distinct epithelial structures need to move toward one another, contact with each other, then allow the cells to reorient themselves in certain polarity, and form a newly fused airway epithelial tube. During this fusion process, the basal cells in the epithelial structures can contact each other at the fusion site, while the apical polarity is maintained (i.e., those differentiating or differentiated cells still locate at the apical layers); meanwhile, the ECM (like collagen, laminin, etc.) around the fusion partners

will be degraded, and the epithelial cells continue to differentiate, forming a larger scale epithelium. The schematic of the airway epithelial fusion events is shown below (Figure 1).

[REDACTED]

Figure 1. Schematic of the changes in the airway epithelium that occur during fusion during avian lung development. The image is cited from the article: Fusion of airways during avian lung development constitutes a novel mechanism for the formation of continuous lumen in multicellular epithelia. *Dev Dyn.* 2020;249(11):1318-1333.

Based on the live videos and microscopic images in the current study, we observe that each epithelial progenitor cell can replicate and proliferate into branch structures; due to the gel is intact during proliferation stage, most branch structures stay separately. Interestingly, when the culture condition is changed to differentiation medium, organoids' branched structures appear to coalesce, differentiate and gradually change to spheroid-like structures; following the ECM degradation, the basal cells from different epithelial structures can contact each other (basal-to-basal docking pattern), and then the merge into larger spheroid-like structures (Figure 2). Therefore, we consider that the merging process of spheroid-like structures in the current organoid system is consistent with the large-scale epithelial fusion events found *in vivo* epithelial airway generation. We have added the relevant content in the Results (**Page 9, Line 182-185**), Discussion (**Page 22-23, Line 481-498**), and a **new Figure S3** (in Supporting information) in the revised manuscript.

Figure 2. Representative multiphoton microscopy images show merged spheroid-like structure organoids in late stages of differentiation cultures. β IV-tubulin staining (green) indicates ciliated cells, MUC5AC staining (red) indicates goblet cells, and the staining in grey color indicates gel.

4) The authors state that in Fig 1A clearly demonstrated is the p63 predominance of cell type. However, the image provided does not represent any green p63 staining. Recommend selecting an alternative. Also, the authors do not describe evaluating comprehensively for airway epithelial types (including non-ciliated non-goblet cells, such as ionocytes). The ZO-1 staining in the provided image is blurry and typical epithelial morphology is not apparent. This is not convincing of tight junction staining. Further the authors state that the pattern of beta-tubulin staining suggests apical out ciliation. Videos do support this statement but the image in figure 1 is not of sufficient resolution to be conclusive.

Reply: Thanks for the comment. In Fig 1A, p63 staining is predominant in organoid during the proliferation (D10) and differentiating (D17) stages. We think the unclear p63 staining in the manuscript may be affected by DAPI (blue color) staining. We have adjusted the intensity of DAPI staining and merged with p63 staining; in addition, we provide the images with single channel staining for p63 in parallel with the images with merged channel for your reference (Figure 3A below).

For the other airway epithelial cell types, we have tried to identify club cells and ionocytes in the current organoids. The results show that club cells (SCGB1A1+) can be stained in the organoids (Figure 3B below), while ionocytes (CFTR+FOXI1+) could not

been found. Ionocyte itself is a rare cell type in airway epithelium (about 1%), and it has been mainly found in lung tissues; therefore, the nasal organoids may be hard to detect ionocytes. We have also added the club cell staining results in supplementary data (**New Figure S2, Page 8, Line 178**) in the revised manuscript.

With regard to ZO-1 staining, we have adjusted the signal intensity of the image file and selected representative images to show a clear ZO-1 staining in the organoids (revised. On the hand, we think the blurry images may be due to the low resolution of images, as usually the image file is compressed in PDF file. Please check the figure below (Figure 3C below).

For the β IV-tubulin staining in Figure 1A as well as the others, please check some representative Z-plane images (cross-sectional view) of β IV-tubulin staining in the organoids, which show apical-out ciliation (Figure 3D below). Moreover, we have also supply more representative images to show the ciliated cells locate on apical-out layer in the revised manuscript (**revised Figure 1A, revised Figure 7B & 7C, new Figure S2, new Figure S3, new Figure S9**).

Figure 3 Representative images for staining of p63 (A), SCGB1A1 (B), ZO-1 (C), and β IV-tubulin in organoids.

5) Authors state “degraded collagen may produce bioactive fragments that likely promote organoid differentiation” – this statement is unclear and not supported by results, should be removed or supported by additional data.

Reply: Thanks for the comment. We agree this statement is not supported by the results, and we have deleted it in the revised manuscript (**Page 9, Line 195-196**).

6) For Figure 1, only the collagen experiments are clearly labeled with sample size. Please clarify the sample size for all imaging experiments completed in this figure and all figures. Furthermore, the authors state sample size as # of experiments in the captions, but do not clarify if these are solely experimental replicates or if there are technical replicates also for each experiment.

Reply: Thanks for the comment. In this revision, we have added the description of sample size for all the figures, and also indicate the numbers of biological replicates and technical replicates in the figure legends.

7) Some of the videos have organoid structures that appear to have something inside them. It could be that some are apical-in structures or artifacts of the time-lapse. This should be clarified in a caption or description for the video.

Reply: As some organoids could merge together to form a bigger organoid structure, and the videos are displayed by time-lapse format, it may have some artifacts inside the organoid structures. We have found that before or after the organoids fusion, the beating cilia still appear on the surface. Moreover, in this revision, we have supplied more organoid experiment, and double checked that both single organoid and fusion organoids clearly show the apical-out ciliation pattern (please check Figure 4 below); some figures presented in this revision also display such patterns (e.g., **Figure 7B, Figure S3**). Anyway, we have indicated a caption for the videos in Supporting information (description of Video S2 and Video S3).

Figure 4 Representative images for staining of β IV-tubulin and MUC5AC in the apical surface of merged organoids.

8) Many of the figures reproduce with such small font as to be illegible. All figures must be legible by the average viewer. Even the higher resolution figures had to be increased in size significantly to be legible.

Reply: Thanks for the comment. In this revision, we have adjusted the font size and resolution in Figure 1, Figure 2, Figure 3, Figure 5 and Figure 7, in order to make the content legible to view.

9) The authors do not state if the gene expression of airway epithelia differs in this model compared to apical-in organoids or monolayer cultures, and whether this is or is not consistent with epithelial differentiation in vivo (for example, in an animal model). This is also a critique for later experiments in the organoid models. Are there other experiments in other model types that could lend additional credence that these findings are not unique to the specific hydrogel or experimental conditions? The findings could be at least somewhat replicated in other culture systems, including monolayers, and the authors should comment on whether there are any other studies in other model types that may also be supportive of the findings in this manuscript.

Reply: Thanks for the comment. In this revision, we have used air-liquid interface (ALI) system to induce airway epithelial differentiation. We have analyzed the gene expression profiles and functional characteristics in nasal epithelial proliferation, differentiating, and differentiated stages. The results show that differentially expressed genes (DEGs) among the different differentiation stages in ALI model are highly overlapped (more than 80%) with those found in current apical-out organoid model. Other than those GO terms related to ciliogenesis of airway epithelium are found in both ALI and organoid system, we do identify that the functional terms related to extracellular matrix organization, assembly, and disassembly are also significant in ALI cultures. In turn, we select genes among the ECM functional groups and compare their expression levels between ALI and organoid models. The heatmap shows a similar expression trend of these genes at proliferation, differentiating and differentiated stages between these two culture systems. The above findings show similar molecular characteristics of the epithelial cells between current hANOs and traditional ALI model, suggesting that the presence of biological functions (like ECM assembly/disassembly) is not an artifact of the culture in this CAH gel system. We have added these data in Results (**Page 12-13, Line 254-272**) and **new Figure S5** (Figure 5 below) in the revised manuscript.

Figure 5 Comparison of molecular characteristics between hANO and ALI model. (A) Venn diagram analysis shows the overlapping genes between the two culture systems. (B) Biological process analysis demonstrates the relevant functional terms in different differentiation stages of hANO and ALI cultures. (C) Functional terms related to extracellular matrix (ECM) organization of these two models are listed in parallel. (D) Heatmap demonstrates regulation genes associated with ECM organization in proliferation and differentiation stages of hANO and ALI cultures.

MMP genes expression is also validated in other airway epithelial culture systems, including ALI and Matrigel apical-in model. Following differentiation progression, mRNA levels of MMP7, MMP9, MMP10, and MMP13 are significantly up-regulated in ALI; while except MMP7, the other three MMP genes show an up-regulation trend in Matrigel model. These findings indicate that production of MMP genes in airway

epithelial cells during differentiation is a general phenomenon and it is not induced by the current hydrogel condition. We have added these data in Results (**Page 14, Line 301-307**) and **new Figure S6** (Figure 6 below) in the revised manuscript.

Figure 6 mRNA levels of MMP7, MMP9, MMP10, and MMP13 were analyzed in ALI (A) and Matrigel model (B) at proliferation (D7) and differentiation (D21) stages by qPCR assays.

Finally, we further investigate the functional roles of MMPs on ciliogenesis of nasal epithelium in ALI and Matrigel apical-in system. When applying MMP inhibitor in the cells in ALI and Matrigel model from the initiation of differentiation, the proportion of ciliated cells as well as the signature gene *Foxj1* was significantly decreased compared to the untreated cells. Taking together, these results are in consistent with the presented hydrogel-based apical-out organoids, confirming the important roles of MMPs in airway ciliation process, and highlighting the normal airway epithelial differentiation is dependent on MMP activity. We have added these data in Results (**Page 17, Line 360-376**) and **new Figure S8** (Figure 7 below) in the revised manuscript.

Figure 7 Characterization of ALI cultures and Matrigel-based apical-in organoids treated with MMP inhibitor. (A) Bright field microscopic views of apical-in organoids treated with and without actinonin during differentiation stages. (B) Representative pictures of candidate cellular markers (β IV-tubulin and MUC5AC) of apical-in organoids. (C-D) β IV-tubulin⁺ cells, MUC5AC⁺ cells and mRNA expression of Foxj1 and MUC5AC were compared in actinonin-treated to untreated ALI or apical-in organoids.

10) The authors state that they selected specific MMPs for further evaluation. Please state why the MMPs selected were chosen – current wording in the text suggests that there are other MMPs that may have been altered. The authors should clarify if there are any other known proteins that participate in ECM degradation or remodeling and

if they also evaluated those genes for altered expression during differentiation as well, from RNA-seq data. This would be more comprehensive and inclusive of possible contributors to the ECM degradation, although further evaluation beyond the RNA-seq would not be considered within scope of this manuscript.

Reply: Thanks for the comment. The specific MMPs (MMP7, MMP9, MMP10, and MMP13) selected for evaluation are based on their changes in expression level in RNA-seq data; and these four MMPs rank the top upregulated genes with statistical difference between differentiation (D24 and D17) and proliferation (D10) stages. We have indicated the selection strategy in Results (**Page 13, Line 284-286**) and added a table of detectable MMP genes (**New Table S2**) in the revised manuscript.

Based on the functional analysis of RNA-seq data, we also find other known metalloproteinases genes that participate in ECM degradation during in tissue remodeling, such as adamalysins family (ADAMs and ADAMTS); for example, down-regulation of ADAM8, ADAM15, ADAMTS1, etc., but up-regulation of ADAM11, ADAM12, ADAM19, ADAM28, ADAMTS4, ADAMTS7, ADAMTS8, ADAMTS9, etc. in differentiation stage as compared to proliferation stage (significant gene list in **Table S1**).

In addition, we select genes among the ECM functional groups (GO terms) and compare their expression levels between ALI and current apical-out organoid models (**New Figure S5**). We found that other metalloproteinases like ADAMs or ADAMTS, as well as other ECM related gene families should be possible contributors to the ECM degradation and their roles on airway epithelial development are worth to study in the future. We have added this discussion in the revised manuscript (**Page 26, Line 566-570**).

11) Video S5 is suggested to show no evidence of cilia, however, the video is zoomed in too far to see much of the apical surface. This is not conclusive of no cilia.

Reply: We have checked more organoid staining images and cytospin images as well as the cilia beating videos in actinonin treated organoids; and we find that the

organoids treated with MMP inhibitor still have few cilia structure, but not absence of cilia. But compared to the untreated organoids at the same differentiation time point, actinonin treated organoids clearly show a reduction of ciliation as well as shorter cilia. We have changed the description of cilia patterns in treated organoids in the revised manuscript (**Page 15, Line 325-326**); in addition, we have also selected other representative videos to show cilia patterns in organoids treated with actinonin (please check **new Video S5** in Supporting information).

12) In the discussion the authors report only 3 studies of apical out airway organoids. However, in fact, there are many going back to the 1990s. However, it may be correct that there are a limited number of culture apical out airway organoids in an artificial extracellular matrix. The authors should revise this sentence to reflect accurately the publications of this type of model.

Reply: Thanks for the comment and reminding. Yes, we agree that the former studies go back to the 1990s have indicated a type of free-floating airway epithelial spheroids [*ref* PMID: 9887058; PMID: 10600878]. This self-assembly sphere culture is similar to the recent ECM-free culture system, and both of them can generate apical-out airway organoids. We consider that the current ECM-free system is optimized from the previous old method, which requires extra operation (such as centrifugation of the cells when changing the medium during differentiation period). Nonetheless, the ECM-free apical-out organoid's limitation is reduction of cell diversity such as lack of goblet cells, club cells and basal cells; and no ECM around the cell units also indicates that such system could not mimic the physiological environment in tissues. Only Boekig et al. used a mixed matrix components (mixture of PureCol and Matrigel) to show apical-out organoids, but this system also doesn't have goblet cells. Therefore, in an artificial extracellular matrix, apical-out airway organoids are still limited. We have added more reference regarding ECM-free apical-out model and revised the description for apical-out airway organoids in the discussion (**Page 21-22, Line 456-479**).

13) The authors must add references on page 19 to the discussion where they are describing characteristics of different collagens.

Reply: We thank the reviewer for this comment. We have added citations to the paragraph in discussion (**Page 24-25, Line 528--541**).

14) Methodology: authors state that patients did not have infection in the month prior to sampling. However, they do not describe testing. Was this conclusion based on patient report or some viral/bacterial testing?

Reply: The subjects recruited in the study have been asked by patient report regarding the respiratory infection symptoms (including fever, cough, runny nose, sore throat, chest or nasal congestion, and fatigue); moreover, nasal swabs from each subject were sent for bacterial culture test and nucleic acid amplification tests for SARS-COV-2, and both tests were confirmed negative. We added this description in the Methods in the revised manuscript (**Page 27-28, Line 596-599**).

Reviewer #2 (Remarks to the Author):

Li et al's manuscript, titled "Primary human apical-out nasal organoids reveal an essential role of matrix metalloproteinases in airway epithelial differentiation and remodeling," describes a novel culture method for apical-out oriented three-dimensional spheroids/organoids of the upper airway that can differentiate into all known major cell types of the mucociliary tract. The authors demonstrate that perturbation of matrix metalloproteinases (MMPs) can alter cellular differentiation.

However, while the story is well written and concise, the manuscript may have limited impact on the field of airway organoids in its current form. Specifically, the manuscript lacks fundamental insights into the presented mechanism of MMPs and cellular differentiation, and the translational value of the results is not well explored or explained, thereby limiting its impact. In conclusion, the manuscript appears to be more technology-based than biology-based. Therefore, it is important for the authors to provide a more detailed discussion on the potential applications of this technology

in order to demonstrate its relevance and impact on the field of airway organoids.

Reply : Thanks for the comment. We agree that the detail mechanism of MMPs underlying airway epithelial cell differentiation still need more experiment to clarify. The current study is aim to present a novel apical-out airway organoid by using a defined hybrid hydrogel system; and by utilizing this system, we have revealed MMPs are required for airway epithelial ciliation, which is associated with the ECM degradation. In this revision, we have added experiments and analysis to compare the findings of current apical-out model with other conventional airway 3D culture system, including air-liquid interface and Matrigel based apical-out model; and we also explore the roles of MMP7 and MMP10 on nasal organoid development; finally, we have discussed more detail about the advantages of current hANOs on recapitulating aspects of cellular complexity and structural characteristics of native nasal epithelium, as well as the potential applications on respiratory research. Please check the point by point response as bellow.

Major points

1. The authors introduce a new hydrogel-based technology for generating apical-out organoids. However, a previous method for generating apical-out organoids has been reported by Stroulios et al. While the authors briefly discuss the comparison between the two methods, a direct comparison of their efficiency in differentiation would greatly benefit the manuscript.

Reply : Thanks for the comment. Recently several studies have reported that apical-out airway organoids can be generated by extracellular matrix (ECM)-free system [Stroulios et al., Sci Reports 2022; Wijesekara et al., 2022]. Such model presents a ciliated cell predominance phenotype, but fails to differentiate goblet cells and also doesn't contain basal cells or club cells. The absence of goblet cells, basal cells and club cells suggests the ECM-free apical-out organoids could not fully represent the cellular composition of native epithelium.

We have checked the technical datasheet of ECM-free organoid system sold by STEMCELL Technologies Ltd, and it introduces organoid formation efficiency to indicate

the differentiation efficiency, i.e., the number of differentiated organoids among every 100 epithelial progenitor cells. As the differentiated organoids in ECM-free system are suspended without fusion, the number of organoids can be easily counted. However, in our culture system, the organoids can merge via the area of basal cells to form larger spheroid-like epithelial structures during differentiation, thus it is hard to calculate the number of organoids for formation efficiency. In our internal data, we have calculated a percentage based on ciliated cell number at D24 divided by progenitor cell number at D10 (i.e., the initiation of differentiation) and this percentage may indicate the differentiation potential. The mean percentage of differentiation potential is 23% at D24.

Nonetheless, we consider that cellular composition is the major difference between current hydrogel-based apical-out organoids and those ECM-free organoids, and we discuss it in detail in the revised manuscript (**Page 21-22, Line 456-479**).

2. The authors utilized human nasal epithelial progenitor cells (hNEPCs) to generate the spheroids. However, it is unclear whether each of the 17 patients was used individually or if a pooled fraction was used, which raises concerns about the efficiency and robustness of the culture method. The manuscript lacks statistics on the number of lines generated from the 17 donors, and as a result, the findings may be interpreted as being based on a single hANO line. To identify common mechanisms and robustness of the system, the results should be repeated with different lines from different donors. **Reply:** Thanks for the comment. The finding of this study is not based on a single hANO line. We used hNEPCs from individual subjects to generate the spheroids, and all hNEPCs from different donors can successfully differentiate into apical-out organoids. In this revision, we have also used nasal brushing to obtain epithelial cells from 4 subjects and then generate differentiated organoids. We have described the number of biological replicates and the experiment number in each figure legend.

3. The authors emphasize the significance of apical-out organoids in the mechanisms described, including the differentiation toward ciliated cells and the shift toward

goblet cell differentiation upon MMP inhibition. To demonstrate the importance of apical-out organoids, the authors could compare their system with conventional apical-in organoids in Matrigel to highlight the added value of their approach. The absence of such a comparison makes it challenging to assess the value of the presented system.

Reply: Thanks for the suggestion. In this revision, we have used two conventional 3D cultures to compare the main findings of the presented system, including air-liquid interface (ALI) models and Matrigel based apical-in organoids. The RNAseq results from ALI system showed a high overlap of differentially expressed genes among differentiation stages with those found in current apical-out organoids (**New Figure S5**). Other than those Gene Ontology (GO) term related to ciliogenesis of airway epithelium, we do identify that the GO terms related to extracellular matrix organization, assembly, and disassembly are also significant functional terms in ALI cultures; moreover, in both organoid and ALI systems, groups of ECM related genes were significantly changed in a similar trend at proliferation and differentiation status (**Page 12-13, Line 254-272 and new Figure S5**). The above findings show similar molecular characteristics of the epithelial cells between current hANOs and traditional ALI model, suggesting that the presence of biological functions (like ECM assembly/disassembly) is not an artifact of the culture in this CAH gel system.

With regards to MMP genes (MMP7, MMP9, MMP10, and MMP13), both ALI and Matrigel system show an increase trend in MMP expression levels following the differentiation (**Page 14, Line 301-307 and new Figure S6**). Importantly, when applied MMP inhibitor (actinonin) in the cells in ALI and Matrigel system, the ciliated cells are significantly suppressed, while the goblet cell proportion seems not change (**Page 17, Line 360-376; New Figure S8**). These results are in consistent with the presented hydrogel-based apical-out organoids, confirming the important roles of MMPs in airway ciliation progression. Thus, these results are in consistent with the presented hydrogel-based apical-out organoids, confirming the important roles of MMPs in airway ciliation process, and highlighting the normal airway epithelial differentiation is dependent on MMP activity. We have added above results in the revised manuscript.

4. The authors state that the spheroids represent an epithelium that mimics the in vivo situation. However, in line 161, the authors suggest that the spheroids have no lumen and are apical-out oriented. This contradicts Figure 1A (MUC5AC d24) and 1B, which shows that the organoids have small lumens containing polarized cells, as indicated by the ZO-1 staining. This suggests that the epithelium is double-polarized, with both apical-out and apical-in orientations. It is debatable whether such an epithelium is representative of the airway epithelium.

Additionally, the authors do not address the question of whether the presence of collagen I on the apical side of cells is representative of the in vivo epithelium, which normally has its apical surface oriented toward the air. This issue is further highlighted by the patient data in Figure 6, which shows differences in basement membrane thickness that is basally located and therefore opposite to the apical-out method presented in this study.

Reply : Thanks for the comment. In Line 161, we describe the organoid images captured by bright field microscopy, showing a lack of clear hollow lumen in the current apical-out organoids as compared to those Matrigel-based apical-in lung organoids. Based on the multiphoton microscopic images, we have found that some nasal organoids have a small lumen; and we have carefully checked the images by Z-plane view (cross sectional side), the organoids with a small lumen still show an apical-out ciliation pattern, and inside the lumen surface there is no evidence of cilia structure (please check Figure 8 below). Moreover, we have also supplied more representative images to show the ciliated cells locate on apical-out layer in the revised manuscript (**revised Figure 1A, revised Figure 7B & 7C, new Figure S2, new Figure S3, new Figure S9**). Thus, we confirm that the epithelium in the current organoid model only present apical-out orientations. We have added the description of above finding in the revised manuscript (**Page 8-9, Line 178-180**).

Figure 8 z-plane view of differentiated organoids.

Regarding the presence of collagen, we have adjusted the angles or direction of the 3D multiphoton microscopic image. We observe that in proliferation or differentiating stage, the organoids are covered around by collagen; while in late differentiation stage, the collagen degrades, and the apical side of the organoid with cilia orients toward the air, i.e., the collagen locates at the basal side of the differentiated organoids (please check Figure 9 below).

Figure 9 Representative pictures show the localization of collagen I on sides of organoids.

5. The mechanism presented is currently only correlated to disease and goblet cell differentiation. The manuscript would greatly benefit genetic ablation of specific MMPs to better identify the biological relevance. Moreover, it would greatly benefit the impact of the new culture system.

Reply: Thanks for the comment. We agree that the mechanism of specific MMP need to be investigated. In the first submitted manuscript, we only suppress MMP9 and MMP13 activity to examine their roles on nasal epithelial differentiation dependent on the commercial availability of the specific MMP inhibitor. Since there is no efficient method to directly and stably knockout specific genes in primary airway epithelial cell cultures during differentiation, we have tried to use neutralizing antibody for specific MMP to inhibit the secreted MMP in the organoid model.

In this revision, we have used neutralizing antibody for MMP7 and MMP10 respectively in the organoid culture during the differentiation period (i.e., applying antibody every 2 days from Day 10 to Day 24). The results show that the gel degradation is delayed and the ciliated cell proportion is significantly reduced in treated organoids as compared to the untreated ones (**New Figure 7**). Moreover, we have also tested siRNA method to specific knockdown MMP9 in organoid system. As siRNA is a transit inhibition assay, we treat the cells with siRNA-MMP9 (si-MMP9) and negative control siRNA (si-NC) every 2 days from Day 10 to Day 24 during the progression of differentiation. The knockdown efficiency for MMP9 in organoids at Day 24 is about 33%; while the results still present a delay of gel degradation and reduction of ciliated cell proportion in si-MMP9 treated organoids versus si-NC treated organoids (**New Figure S9**). The goblet cell (Mucin5AC+) percentages seem not affected by the treatment with neutralizing antibodies or siRNA. We have added the above results in the revised manuscript (**Page 19-20, Line 403-433; new Figure 7, Figure S9**).

6. In their supplementary video S2 and S3 there are clearly organoids moving through the viewing field. The authors do not comment on this or explain the significance.

Reply: Thanks for the comment. Videos S2 and S3 show the live view of organoids during differentiation period. The movement of organoids is explained by two reasons: 1) the ECM (collagen) in gel gradually degrades, making some space around organoids; 2) organoids undergo ciliation on the apical side of the epithelial structure and then the beating cilia can drive the movement of organoids in the gel system.

Based on the live videos and the 3D multiphoton microscopic images, we find that some moving organoids during differentiation stages will merge together to form a larger spheroid. Such phenomenon is similar to the findings reported in several developmental studies showing that the continuous network of airway is generated by large-scale epithelial fusion events *in vivo* [ref PMID: 32510705; PMID: 35800889]. When checking the 3D immunofluorescent images, we also find that two spheroids merge via the site with basal or progenitor cells, while the site with differentiated cells (like ciliated cells) in organoids maintains intact and doesn't fuse (Please check the Figure 4 above in Reviewer 1's comment). We have also added the above content in Results (**Page 9, Line 182-185, new Figure S3**) and Discussion (**Page 22-23, Line 481-498**) in the revised manuscript.

Minor points

1. In line 233, the authors state that they examined all MMP genes, but they do not provide this data. Including this information in the manuscript would allow the authors to provide stronger evidence for the biological significance of the MMPs they eventually highlight. Additionally, it would be advantageous to investigate this in various lines from different donors (see major comment 2)

Reply: Thanks for the comment. We have examined the expression levels of MMP genes based on RNA-seq data. Here we provide a table to show the changes (fold-change and statistical difference) of MMP genes among differentiation versus proliferation (**new Table S2**). We add this description in revised manuscript (**Page 13, Line 283-285**). The RNA-seq experiment in Figure 2 is performed in the three lines of nasal organoids from three different donors, and the qPCR and Luminex experiment in Figure 3 are performed in four to five lines of nasal organoids from different donors. We have indicated the numbers in Figure legends.

2. Figure 1D shows that the highest MMP activity is observed in day 10 organoids, which consist of proliferating basal cells. This result is unexpected, yet the authors do not offer any commentary or explanation on it.

Reply: Thanks for the comment. The MMP activity assay kit uses a fluorescence

resonance energy transfer (FRET) peptide as a generic MMP activity indicator. The kit is designed to measure general activity of MMP enzymes, and it doesn't give an individual read-out for each MMP. In RNA-seq data, we also find that some MMP genes are up-regulated on Day 10 versus Day 17 or Day 24 (such as MMP14 and MMP1), which may contribute to the total MMP activity at proliferation stage.

During proliferation status, epithelial progenitors can proliferate and migrate into branched structures, in which this cellular process should need to regulate the ECM assembly/disassembly progression. Therefore, the total MMP activity may be still high during proliferation. Nonetheless, our results show when the organoids initiate differentiation, the total MMP activity is gradually enhanced with up-regulation of certain MMP genes in D24 versus D17. It is worth to investigate the roles of different MMP genes at proliferation and differentiation progression respectively in the future, which can provide a comprehensive view for the functions of different MMP family genes on the airway epithelial development. We have added some of the above content in the discussion in revised manuscript (**Page 25-26, Line 548-555**).

3. Line 255-258 claims differences in morphology in the organoids. While the brightfield images are convincing, the authors could strengthen their point by quantifying the circularity of the organoids.

Reply: Thanks for the suggestion. We have calculated the percentages of circularity of the organoids among total organoids in samples treated with MMP inhibitor versus untreated organoids. As on Day 24, most organoids merge together and it is hard to count these merged spheroids; thus we only quantify the organoid numbers on Day 17. The results show a significant reduction of circular organoids in samples treated with actinonin as compared to the untreated samples (Figure 10 below).

Figure 10 Comparison of circular organoids between organoids treated with and without MMP inhibitor.

4. Video S5 is very short and therefore not relevant to view

Reply: We have carefully checked the organoid videos and have found that there are still several beating cilia in the organoids treated with actinonin. But the ciliary patterns are obvious different, showing that the organoids treated with MMP inhibitor display shorter cilia, smaller cilia area and slower beating frequency as compared to those found in untreated organoids at the same day. These findings are consistent with the immunofluorescence staining in cytopsin slides, which still show fewer ciliated cells in actinonin treated organoids. Therefore, we consider that MMP inhibition can suppress the ciliation progression of organoids, but not completely inhibit ciliated cell differentiation. In the revised manuscript, we have modified the description of the ciliary patterns in actinonin treated organoids in Results part (**Page 15, Line 320-322**), and also replaced new representative videos to show the ciliary beating patterns in actinonin treated organoids (**New Video S5** in Supporting information).

5. The mechanism behind MMP9 inhibition and its association with goblet cell differentiation remains unexplored in this manuscript. If the authors can provide more insight into this mechanism, it would enhance the biological relevance of their findings. For instance, is this solely due to the spatial requirements for generating cilia, or is there a signaling cascade that is altered, leading to differentiation towards an alternative pathway? There are various studies that have been conducted on the

differences of ciliated and goblet cell differentiation, which the authors could examine to identify potential mechanisms.

Reply: Thanks for the comment. We haven't explored the potential mechanism of MMP9 inhibition on organoid cultures. But the differentiation phenotypes are consistent in actinonin treated organoids with MMP9 inhibitor treated organoids, i.e., both of these two inhibition assays show a decrease of ciliated cell proportion while enhance of goblet cell percentage. Hereby, we suppose that pan-MMP inhibition on epithelial differentiation may share similar mechanisms with MMP9 inhibition.

Based on the RNA-seq analysis, we have characterized the molecular profiles in organoids treated with MMP inhibitor versus those that were untreated. Other than the spatial requirements for generating cilia, several important genes or signaling cascades on ciliated cell or goblet cell differentiation have been identified. For instance, IL-13 and IL1B are up-regulated in actinonin treated organoids, in which these genes are known factors to induce goblet cell differentiation. The analysis showed that signaling pathways like Wnt signaling pathway, IL-17 signaling pathway, ECM-receptor interactions are also significant altered in organoids treated with MMP inhibitors, in which these pathways are important functional process to regulate airway epithelial differentiation. Therefore, epithelial derived MMP – ECM assembly/disassembly – signaling cascades in epithelial cells may be an important functional axis to regulate airway epithelial differentiation, which is required to further investigate in detail.

6. All heatmaps presented have no clear legend that indicates which values are presented. The colour scale is thereby imbalanced around -0.5 instead of 0.

Reply: Thanks for the comment. We have added the explanation of the values of heatmap in legends of Figure 3 and 5, and also adjusted the color scale presentation in these two figures (**revised Figure 3 and revised Figure 5**)

7. In figure 3B-C, the authors use line graphs to indicate differences in expression. This however pretends that the same culture was followed over time and time points were

isolated from the exact same organoids. This is not feasible. The authors should therefore replace the line graph with bar graphs and statistics.

Reply: Thanks for the comment. In Figure 3B-C, we have used line graphs to indicate the differences of MMP expression levels following differentiation time points. Each line represents an organoid from different donor. Since the expression levels of MMPs at Day 10 are very low, most of the data points are stacked together in the lower end of Y-axis, making the lines unclear to differentiate. In the revised version, we have performed log transform in the scale of Y-axis, in order to clearly show the expression levels of MMPs from different organoid samples (**Revised Figure 3**). We have also indicated the biological and technical replicates in Figure legend.

Textual/visual points

1. Line 49 has spelling error “to access” should be “accessing the”
2. Line 51 misses commas for easier reading
3. Line 93 misses “a” before murine tumor
4. The word system in line 94 is not needed
5. Line 99-100: “... cells in Matrigel more tend to grow inwards...” should be “... cells in Matrigel tend to grow more inwards...”
6. Line 110 has one capital too many
7. Line 111 has one space too many
8. Line 196 has spelling error in “genes”
9. Figure 1D is very small

Reply: Thanks for pointing out the above typos and mistakes. We have corrected them in the revised manuscript following the suggestion.

Reviewer #3 (Remarks to the Author):

This manuscript describes the preparation of a biochemically-defined hybrid gel comprised of collagen type I, alginate, and hyaluronic acid. Human nasal epithelial progenitor cells embedded in this gel degrade the ECM to create hollow spaces within which the cells differentiate into ciliated cells. Addition of protease inhibitors suppress

ECM degradation, reduce ciliation and promote mucus-secreting cell production.

Gene expression studies show increase in matrix metalloproteinases (MMP7, MMP9, MMP10 and MMP13) during differentiation into ciliated cells. Moreover, a decrease of MMPs was found in goblet cell hyperplastic epithelium obtained from patient tissue. The patient tissue histology is very nice.

1. While the novel material system is interesting and impacts nasal cell differentiation, how well the observations recapitulate development is not clear and not convincing since alginate is not very physiological and presence of type I collagen and hyaluronic acid but not other ECM or basement membrane proteins also make the system artificial. It may be better to simply state that the materials system is interesting for producing a type of apical out organoid without suggesting similarity to development.

Reply : Thanks for the comment. The addition of alginate helps to maintain the integrity of the gel system; while type I collagen and hyaluronic acid are major ECM components in mucosal tissue. We agree that the current hydrogel system could not fully represent the composition of ECM in airway tissues, as ECM components are very complex. We have prevented making overstatement about the similarity of current hydrogel system with ECM environment in tissues.

Nonetheless, in the current organoids, we have detected basal cells, ciliated cells, goblet cells and club cells, in which the cellular composition is more diverse than those apical-out organoids in ECM free system (no goblet cells). In addition, we have observed branched structures during proliferation stage, and fusion events of epithelial spheroids during differentiation stage; this above phenomenon could not be found in Matrigel system, and it may represent certain step in epithelial development *in vivo*. We have added this discussion in the revised manuscript (**Page 21-22, Line 456-479; Page 22-23, Line 481-498**).

2. The method is advantageous to some other apical-out airway organoids as there is no need to manually remove ECM. On the other hand, minimal ECM method also exist that avoid this difficulty. Easy pathogen accessibility is a potential advantage to be

demonstrated. But several other apical out airway organoids reported may also have the same advantage.

Reply: Thanks for the comment. Currently, apical-out airway organoids are mainly generated by ECM-free system, as well as those Matrigel based system which needs to manually remove ECM. Regarding the pathogen accessibility, the current hANOs is similar to the other apical-out organoids reported in recent studies. However, the other apical-out organoids, especially those generated from ECM-free system could not differentiate to goblet cell, which is the main functional cell in epithelium responding to pathogen stimulations. Hence, the lack of cell type diversity in other apical-out airway organoids may limit their applications in studying epithelial development *in vitro*, and the host response to pathogen stimulation. We have added this discussion in the revised manuscript (**Page 21-22, Line 456-479**)

3. Page 9 the comment “degraded collagen may produce bioactive fragments that likely promote organoid differentiation”. is a bit hollow. That is this is hard to appreciate, without some reference, analysis or other evidence to back up the comment.

Reply: Thanks for the comment. We agree this description is over interpreted, and have deleted it (**Page 9, Line195-196**).

4. Page 9 “system could morphologically and functionally recapitulate the physiological features of human upper airway epithelium.” In what way is the author suggesting the system is morphologically physiological? Isn't the morphology opposite of physiological?

Reply: Thanks for the comment. We have changed the description to “...which this system could recapitulate the 3D structural characteristics and the cellular complexity of human nasal epithelium.” (**Page 9, Line 199-200**).

5. Is MMP upregulation during ciliogenesis an artefact of the culture in the engineered hydrogels or a physiological reality and necessity? MMP inhibitor addition leads to a more “branching” type of morphology. But is this really branching or just attachment

to the ECM of fragments in various shapes due to just attachment to the hydrogel? More evidence of actual branching would be needed to make these claims.

Reply: Thanks for the comment. In this revision, we have performed air-liquid interface (ALI) and Matrigel-based organoid cultures. Interestingly, we have also found that MMP genes (MMP7, MMP9, MMP10 and MMP13) are up-regulated during differentiation progress (**Page 14, Line 301-307, new Figure S6**); when applying MMP inhibitor in these culture models, the ciliation progress has been inhibited (**Page 17, Line 360-376, new Figure S8**). Therefore, the upregulation of MMP genes during ciliogenesis is not an artefact of the culture; and the current organoid should be an idea model to study the interaction between MMP and ECM (like collagen) assembly/disassembly process during airway epithelial differentiation.

With regard to the “branching” type of morphology during proliferation stage, based on the 3D multiphoton microscopic images, p63+ progenitor cells arrange in a branch-like structure on Day 10 (**Figure 1**). By viewing the organoids in live videos (**Video S1 and S2 in Supporting information**), we observe that the epithelial progenitors can proliferate and migrate into branching shape in 3D cultures during proliferation stage; when treating the organoids with MMP inhibitor, the differentiation progression of organoids is suppressed, and then the organoids could not change to spheroid-like structure, therefore, they are “stuck” in branching morphology. Hence, the above evidence indicates that the organoids grow in an actual branching structure during proliferation stage.

6. Page 17 “better recapitulate the biological properties of native airway epithelial epithelium, from proliferation to differentiation” what the basis is for the authors to conclude this is not clear. What specific findings are you pointing to when you make this claim? In particular, some of the findings, such as disappearance of basal cell types near completely from the organoids suggest the cultures may not be physiological.

Reply: Thanks for the comment. We agree that the current organoids may not fully recapitulate all physiological features of nasal epithelium *in vivo*; however, this nasal organoid can still recapitulate certain characteristics of epithelial development in

native epithelium from human nasal mucosa, including epithelial morphological structure, cellular composition, and intrinsic property of growth/differentiation patterns. This claim is based on the following evidence:

1) the branching morphogenesis in 3D organoids during proliferation stage may mimic the morphological patterns of epithelial cell proliferation and migration *in vivo* tissues (**Figure 1**);

2) during differentiation, the branched organoids are gradually transformed to spheroid-like organoids, and merge into larger scale of epithelial structures, in which this fusion event is an essential step during epithelial development in many types of tissues (like lung, skin, and gut) [*ref* PMID: 9887058; PMID: 10600878]; We have added the relevant content in the Results (**Page 9, Line 182-185**), Discussion (**Page 22-23, Line 481-498**), and a **new Figure S3** (in Supporting information) in the revised manuscript.

3) at proliferation stage, the organoids are mainly composed of p63+ basal cells, following differentiation, percentage of basal cells is decreased, while percentage of differentiated cell types (ciliated cells, goblet cells) are increased, moreover, club cells appear and a few of basal cells still maintain in the differentiated organoids (Note: we have adjusted the brightness and contrast of the multiphoton microscopy images (**Figure 1**); the results show a clear basal cell staining on D24 although its percentage is lower than Day 10 or Day 17; **Figure 11** below);

Figure 11 p63+ basal cell prevalence in hANOs in D10, D17, and D24.

4) as airway epithelial progenitor cells are considered to maintain certain inherent properties of growth and differentiation procession from the native epithelium. We have added the relevant content in Discussion in the revised manuscript (**Page 23-24, Line 500-514**).

Nonetheless, we have revised the description related to the term “physiological” and added the above content in Discussion part in revised manuscript.

7. Page 18. Also, while the authors state ref 21,22, 25 are not physiological for not having goblet cells, the current system losing almost all of their basal cells is also not physiological.

Reply: Because when the epithelial cells applying differentiation medium, they will differentiate into ciliated cells or goblet cells and can't turn back to basal cells. Thus,

all the *in vitro* airway epithelial culture system, including organoids and air-liquid interface models, show a reduction of basal cell proportion in late differentiation stage; on the other hand, the differentiated nasal organoids should contain differentiated cell types, including ciliated cells and goblet cells, as well as few club cells. Therefore, we claim that compared to those apical-out organoids reported in the literature, the current organoids are better to represent the cellular composition of native nasal epithelium.

As explained in the No.6 comment, we do observe a few p63+ basal cells inside the organoids on late differentiation stage. We have adjusted the brightness and contrast of the multiphoton microscopy images (**Figure 1**) and the results still show a clear basal cell staining on Day 24.

8. Ref 44-46 suggest artificial environment promote MMPs. Could the situation here be just the same? The authors do show differences in MMP in goblet hyperplasia later in life in patient tissue samples but whether MMP is involved during DEVELOPMENT as state by authors is not definitive.

Reply: Thanks for the comment. Refs 44-46 show that MMPs are induced in airway epithelial cultures by external stimulants like ceramide, house dust mite and formaldehyde, indicating the roles of epithelial-derived MMPs underlying environmental factors mediated cellular damages. During this revision period, we have added air-liquid interface and Matrigel-based organoid culture experiment. Both of these two systems show enhance of MMP expression following differentiation progression, while inhibition of MMP activity can suppress ciliation in the culture models, which is same as the current apical-out model. We have added these results in Results section (**Page 14, Line 301-307, and new Figure S6; Page 17, Line 360-376, and new Figure S8**) in the revised manuscript. Therefore, we consider that airway epithelial cells can produce MMP without external stimulants during the differentiation, which is essential for normal ciliation in epithelium.

9. Several claims such as role of collagen fragments, role of MMPs is by the authors

own statements still speculative.

Reply : Thanks for the comment. Some description about the roles of collagen fragments have been deleted in the Results part (**such as, Page 9, line 195-196**). We have also reorganized the description about the collagen fragments in Discussion part (**Page 26, Line 557-560**), and some statements which are over interpreted have been deleted (**Page 26, Line 560**).

10. Schematic in Fig 8 is a misrepresentation. The authors histology and text say there is hardly any basal cells (very little p63) and no lumen.

Reply : Thanks for the comment. Indeed, when checking the multiphoton microscopic images via the Z-plane view (the organoids viewing in cross sectional side), there is still a small lumen inside the organoid (like **Figure 1A, Figure 7B & C, Figure S3**); while in 3D construction or Maximum Z-stack view, the lumen could be masked by the overlaid planes. In this revision, we have revised the description about the lumen in Results part (**Page 8-9, Line 178-180**) and depict the lumen in as smaller size in Schematic picture (**Figure 8**). With regards to the presentation of basal cells, we reduce the number of basal cells and just draw one layer of basal cells in the organoid. Please check the revised schematic in Figure 8.

11. In summary, the description of apical out nasal organoids is interesting. And the hydrogel material may be somewhat novel. MMP differences in normal vs diseased patient samples is convincing and supported by beautiful histology.

The authors are inaccurate or overly speculative in their analysis, description, and depiction (for example **Figure 8**).

Reply : Following the reviewer's comment, we have made revision about the statement which is inaccurate or overly speculative (please check the above point-by-point response). We appreciate the constructive comment which improves the manuscript.

REVIEWERS' COMMENTS

Reviewer #1 (Remarks to the Author):

The authors have completed additional experiments and have been majorly responsive to all critiques, satisfactorily revising the manuscript. No further revisions are requested from this reviewer.

Reviewer #2 (Remarks to the Author):

The revised manuscript of Li et al has significantly improved and has led to increased impact of the data shown as well as the model presented. I thank the authors for their in-depth rebuttal and revision. Overall, all comments raised have been answered in a significant manner. Below, a point-by-point reply highlights the changes.

Major points

1. The authors introduce a new hydrogel-based technology for generating apical-out organoids. However, a previous method for generating apical-out organoids has been reported by Stroulios et al. While the authors briefly discuss the comparison between the two methods, a direct comparison of their efficiency in differentiation would greatly benefit the manuscript.
2nd Reply: The authors now included in-depth hypothetical comparison between existing models and their novel system. While experimental comparison lacks for some existing models, the authors clearly state in their discussion the benefit of the current model over the older models which increases the relevance of the manuscript.
2. The authors utilized human nasal epithelial progenitor cells (hNEPCs) to generate the spheroids. However, it is unclear whether each of the 17 patients was used individually or if a pooled fraction was used, which raises concerns about the efficiency and robustness of the culture method. The manuscript lacks statistics on the number of lines generated from the 17 donors, and as a result, the findings may be interpreted as being based on a single hANO line. To identify common mechanisms and robustness of the system, the results should be repeated with different lines from different donors.
2nd Reply : Addition of the data on biological replicates in the figure legend has helped the understanding of the use of different donors to show robustness of the system. Minor additional detail: the authors could include their efficacy of generating apical-out organoids in the text to inform readers of this robustness even more.
3. The authors emphasize the significance of apical-out organoids in the mechanisms described, including the differentiation toward ciliated cells and the shift toward goblet cell differentiation upon MMP inhibition. To demonstrate the importance of apical-out organoids, the authors could compare their system with conventional apical-in organoids in Matrigel to highlight the added value of their approach. The absence of such a comparison makes it challenging to assess the value of the presented system.

2nd Reply : The addition of comparison with conventional ALI cultures using RNAseq greatly increased the significance of their finding. The new model follows expected lineage commitment but also now shows novelty with the MMP signaling.

4. The authors state that the spheroids represent an epithelium that mimics the in vivo situation. However, in line 161, the authors suggest that the spheroids have no lumen and are apical-out oriented. This contradicts Figure 1A (MUC5AC d24) and 1B, which shows that the organoids have small lumens containing polarized cells, as indicated by the ZO-1 staining. This suggests that the epithelium is double-polarized, with both apical-out and apical-in orientations. It is debatable whether such an epithelium is representative of the airway epithelium.

Additionally, the authors do not address the question of whether the presence of collagen I on the apical side of cells is representative of the in vivo epithelium, which normally has its apical surface oriented toward the air. This issue is further highlighted by the patient data in Figure 6, which shows differences in basement membrane thickness that is basally located and therefore opposite to the apical-out method presented in this study.

2nd Reply : The images provided are more visual on the apical-out morphology of the organoids. This clearly indicates the polarization of the organoids. Additionally, by adding the lines about the potential smaller lumens inside the organoids, the authors have shown full transparency on the process.

5. The mechanism presented is currently only correlated to disease and goblet cell differentiation. The manuscript would greatly benefit genetic ablation of specific MMPs to better identify the biological relevance. Moreover, it would greatly benefit the impact of the new culture system.

2nd Reply : The additional data more clearly indicates the relevance of the MMPs. While genetic ablations would still strengthen their point, the authors have now provided enough evidence for a clear correlation between the observed phenomena.

6. In their supplementary video S2 and S3 there are clearly organoids moving through the viewing field. The authors do not comment on this or explain the significance.

2nd Reply : By explaining the observed movement as well as its relevance in the text, the authors have now covered the issue.

Minor points

1. In line 233, the authors state that they examined all MMP genes, but they do not provide this data. Including this information in the manuscript would allow the authors to provide stronger evidence for the biological significance of the MMPs they eventually highlight. Additionally, it would be advantageous to investigate this in various lines from different donors (see major comment 2)

2nd Reply : The data on all MMP genes is now added and the different donors issue is solved by addition of the biological replicates in the figure legends.

2. Figure 1D shows that the highest MMP activity is observed in day 10 organoids, which consist of proliferating basal cells. This result is unexpected, yet the authors do not offer any commentary or explanation on it.

2nd Reply : Additional explanation is added by the authors

3. Line 255-258 claims differences in morphology in the organoids. While the brightfield images are convincing, the authors could strengthen their point by quantifying the circularity of the organoids.

2nd Reply : Figure 10 was added and this helps the visualization into quantification and thereby strengthens the author's claim.

4. Video S5 is very short and therefore not relevant to view

2nd Reply : We thank the authors for the more thorough explanation of the observed ciliary beating. The new video S5 however remains short (2sec). This makes the visualization difficult. Perhaps a slow-motion version can be added to have a longer video to look at the ciliary beating. This is a simply aesthetic, not scientific comment.

5. The mechanism behind MMP9 inhibition and its association with goblet cell differentiation remains unexplored in this manuscript. If the authors can provide more insight into this mechanism, it would enhance the biological relevance of their findings. For instance, is this solely due to the spatial requirements for generating cilia, or is there a signaling cascade that is altered, leading to differentiation towards an alternative pathway? There are various studies that have been conducted on the differences of ciliated and goblet cell differentiation, which the authors could examine to identify potential mechanisms.

2nd Reply : Many thanks to the authors for their explanation and reasoning. Addition of these thoughts in the discussion might inspire as well.

6. All heatmaps presented have no clear legend that indicates which values are presented. The colour scale is thereby imbalanced around -0.5 instead of 0.

2nd Reply : Legends have been changed properly.

7. In figure 3B-C, the authors use line graphs to indicate differences in expression. This however pretends that the same culture was followed over time and time points were isolated from the exact same organoids. This is not feasible. The authors should therefore replace the line graph with bar graphs and statistics.

2nd Reply : By explaining that each line is a different donor, the problem is solved by the authors

Reviewer #3 (Remarks to the Author):

Overall an interesting paper where studies with a novel in vitro model combined with analysis of donor tissues reveal a role of MMPs in disease. The added information and edits are useful.

An added bonus that the organoids formed have an interesting apical-out form which may have additional interesting uses.

The title, however, deviates from actual content and message of the paper. The investigators reveal an essential role of MMP in airway epithelial cell differentiation using CAH. And the CAH system used to study this happens to produce apical-out organoids. But the apical-out organoids are not needed to reveal this as demonstrated by the authors themselves using other systems:

“production of MMP genes in airway epithelial cells during differentiation is a general phenomenon and it is not induced by the current hydrogel condition” also “The ciliation process of airway epithelium is dependent on MMP activity in ALI and apical-in organoid model”

There are applications where the apical-out would be beneficial but analyzing the role of MMP is not one of them.

Demonstrating MMP differences between normal versus goblet cell hyperplasia tissue is useful.

Gene expression pattern comparison between ALI and Matrigel and hANO is useful.

SI video 1 is still too fast or doesn't include any meaningful movie.

REVIEWER COMMENTS

Reviewer #1 (Remarks to the Author):

The authors have completed additional experiments and have been majorly responsive to all critiques, satisfactorily revising the manuscript. No further revisions are requested from this reviewer.

Reply: Thanks for the constructive comment and suggestion for improving the manuscript.

Reviewer #2 (Remarks to the Author):

The revised manuscript of Li et al has significantly improved and has led to increased impact of the data shown as well as the model presented. I thank the authors for their in-depth rebuttal and revision. Overall, all comments raised have been answered in a significant manner. Below, a point-by-point reply highlights the changes.

Reply: Thanks for the constructive comment and suggestion for improving the manuscript.

Major points

1. The authors introduce a new hydrogel-based technology for generating apical-out organoids. However, a previous method for generating apical-out organoids has been reported by Stroulios et al. While the authors briefly discuss the comparison between the two methods, a direct comparison of their efficiency in differentiation would greatly benefit the manuscript.

2nd Reply: The authors now included in-depth hypothetical comparison between existing models and their novel system. While experimental comparison lacks for some existing models, the authors clearly state in their discussion the benefit of the current model over the older models which increases the relevance of the manuscript.

2. The authors utilized human nasal epithelial progenitor cells (hNEPCs) to generate the spheroids. However, it is unclear whether each of the 17 patients was used individually or if a pooled fraction was used, which raises concerns about the efficiency and robustness of the culture method. The manuscript lacks statistics on the number of lines generated from the 17 donors, and as a result, the findings may be interpreted as being based on a single hANO line. To identify common mechanisms and robustness of the system, the results should be repeated with different lines from different donors.

2nd Reply: Addition of the data on biological replicates in the figure legend has helped the understanding of the use of different donors to show robustness of the system.

Minor additional detail: the authors could include their efficacy of generating apical-out organoids in the text to inform readers of this robustness even more.

Author reply: Thanks for the comment. We have added a description for the efficacy of generating apical-out organoids in the Results section (**Page 8, Line 162-164**).

3. The authors emphasize the significance of apical-out organoids in the mechanisms described, including the differentiation toward ciliated cells and the shift toward goblet cell differentiation upon MMP inhibition. To demonstrate the importance of apical-out organoids, the authors could compare their system with conventional apical-in organoids in Matrigel to highlight the added value of their approach. The absence of such a comparison makes it challenging to assess the value of the presented system.

2nd Reply: The addition of comparison with conventional ALI cultures using RNAseq greatly increased the significance of their finding. The new model follows expected lineage commitment but also now shows novelty with the MMP signaling.

4. The authors state that the spheroids represent an epithelium that mimics the in vivo situation. However, in line 161, the authors suggest that the spheroids have no lumen and are apical-out oriented. This contradicts Figure 1A (MUC5AC d24) and 1B, which shows that the organoids have small lumens containing polarized cells, as indicated by

the ZO-1 staining. This suggests that the epithelium is double-polarized, with both apical-out and apical-in orientations. It is debatable whether such an epithelium is representative of the airway epithelium.

Additionally, the authors do not address the question of whether the presence of collagen I on the apical side of cells is representative of the in vivo epithelium, which normally has its apical surface oriented toward the air. This issue is further highlighted by the patient data in Figure 6, which shows differences in basement membrane thickness that is basally located and therefore opposite to the apical-out method presented in this study.

2nd Reply: The images provided are more visual on the apical-out morphology of the organoids. This clearly indicates the polarization of the organoids. Additionally, by adding the lines about the potential smaller lumens inside the organoids, the authors have shown full transparency on the process.

5. The mechanism presented is currently only correlated to disease and goblet cell differentiation. The manuscript would greatly benefit genetic ablation of specific MMPs to better identify the biological relevance. Moreover, it would greatly benefit the impact of the new culture system.

2nd Reply: The additional data more clearly indicates the relevance of the MMPs. While genetic ablations would still strengthen their point, the authors have now provided enough evidence for a clear correlation between the observed phenomena.

6. In their supplementary video S2 and S3 there are clearly organoids moving through the viewing field. The authors do not comment on this or explain the significance.

2nd Reply: By explaining the observed movement as well as its relevance in the text, the authors have now covered the issue.

Minor points

1. In line 233, the authors state that they examined all MMP genes, but they do not provide this data. Including this information in the manuscript would allow the authors to provide stronger evidence for the biological significance of the MMPs they eventually highlight. Additionally, it would be advantageous to investigate this in various lines from different donors (see major comment 2)

2nd Reply: The data on all MMP genes is now added and the different donors issue is solved by addition of the biological replicates in the figure legends.

2. Figure 1D shows that the highest MMP activity is observed in day 10 organoids, which consist of proliferating basal cells. This result is unexpected, yet the authors do not offer any commentary or explanation on it.

2nd Reply: Additional explanation is added by the authors

3. Line 255-258 claims differences in morphology in the organoids. While the brightfield images are convincing, the authors could strengthen their point by quantifying the circularity of the organoids.

2nd Reply: Figure 10 was added and this helps the visualization into quantification and thereby strengthens the author's claim.

4. Video S5 is very short and therefore not relevant to view

2nd Reply: We thank the authors for the more thorough explanation of the observed ciliary beating. The new video S5 however remains short (2sec). This makes the visualization difficult. Perhaps a slow-motion version can be added to have a longer video to look at the ciliary beating. This is a simply aesthetic, not scientific comment.

Author reply: Thanks for the suggestion. The authors have adjusted the video to a slow-motion version (0.5x speed of the original one) and submit it in this revision.

5. The mechanism behind MMP9 inhibition and its association with goblet cell

differentiation remains unexplored in this manuscript. If the authors can provide more insight into this mechanism, it would enhance the biological relevance of their findings. For instance, is this solely due to the spatial requirements for generating cilia, or is there a signaling cascade that is altered, leading to differentiation towards an alternative pathway? There are various studies that have been conducted on the differences of ciliated and goblet cell differentiation, which the authors could examine to identify potential mechanisms.

2nd Reply: Many thanks to the authors for their explanation and reasoning. Addition of these thoughts in the discussion might inspire as well.

Author reply: Thanks for the suggestion. We have added the explanation of potential mechanism underlying the goblet cell differentiation in organoids treated with MMP inhibitors in the Discussion section (**Page 26, Line 557-561**).

6. All heatmaps presented have no clear legend that indicates which values are presented. The colour scale is thereby imbalanced around -0.5 instead of 0.

2nd Reply: Legends have been changed properly.

7. In figure 3B-C, the authors use line graphs to indicate differences in expression. This however pretends that the same culture was followed over time and time points were isolated from the exact same organoids. This is not feasible. The authors should therefore replace the line graph with bar graphs and statistics.

2nd Reply: By explaining that each line is a different donor, the problem is solved by the authors

Reviewer #3 (Remarks to the Author):

Overall an interesting paper where studies with a novel in vitro model combined with analysis of donor tissues reveal a role of MMPs in disease. The added information and edits are useful.

An added bonus that the organoids formed have an interesting apical-out form which may have additional interesting uses.

The title, however, deviates from actual content and message of the paper. The investigators reveal an essential role of MMP in airway epithelial cell differentiation using CAH. And the CAH system used to study this happens to produce apical-out organoids. But the apical-out organoids are not needed to reveal this as demonstrated by the authors themselves using other systems:

“production of MMP genes in airway epithelial cells during differentiation is a general phenomenon and it is not induced by the current hydrogel condition” also “The ciliation process of airway epithelium is dependent on MMP activity in ALI and apical-in organoid model”

There are applications where the apical-out would be beneficial but analyzing the role of MMP is not one of them.

Demonstrating MMP differences between normal versus goblet cell hyperplasia tissue is useful.

Gene expression pattern comparison between ALI and Matrigel and hANO is useful.

SI video 1 is still too fast or doesn't include any meaningful movie.

Author reply: Thanks for the comment and suggestion. Following the reviewers' comment, we have supplied ALI and Matrigel based culture system, and have found that airway epithelial differentiation is dependent on MMP activity. These results strengthen the essential role of MMPs on airway epithelial ciliation. Our findings also demonstrate that degradation of ECM is required to determine the polarity of airway epithelium. Hence, the current apical-out organoid model is useful in studying the interaction between tissue microenvironment factors and ECM organization in regulation of airway epithelial development *in vitro*. We have indicated the above content in both Introduction (e.g., **Page 6, Line 116-119**) and Discussion part (e.g., **Page 26, Line 561-565**).

In addition, the authors have adjusted the Supplementary Video 1 to a slow-motion version (0.75x speed of the original one) and submit it in this revision.